# Emerging MXene–Polymer Hybrid Nanocomposites for High-Performance Ammonia Sensing and Monitoring

**DOI:** 10.3390/nano11102496

**Published:** 2021-09-24

**Authors:** Vishal Chaudhary, Akash Gautam, Yogendra K. Mishra, Ajeet Kaushik

**Affiliations:** 1Research Cell and Department of Physics, Bhagini Nivedita College, University of Delhi, New Delhi 110045, India; 2Centre for Neural and Cognitive Sciences, University of Hyderabad, Hyderabad 500046, India; akash@uohyd.ac.in; 3Mads Clausen Institute, NanoSYD, University of Southern Denmark, Alsion 2, 6400 Sønderborg, Denmark; Mishra@mci.sdu.dk; 4NanoBioTech Laboratory, Health System Engineering, Department of Environmental Engineering, Florida Polytechnic University, Lakeland, FL 33805, USA

**Keywords:** MXene–polymer nanocomposite, ammonia sensing, intelligent sensing device, health wellness, environmental monitoring

## Abstract

Ammonia (NH_3_) is a vital compound in diversified fields, including agriculture, automotive, chemical, food processing, hydrogen production and storage, and biomedical applications. Its extensive industrial use and emission have emerged hazardous to the ecosystem and have raised global public health concerns for monitoring NH_3_ emissions and implementing proper safety strategies. These facts created emergent demand for translational and sustainable approaches to design efficient, affordable, and high-performance compact NH_3_ sensors. Commercially available NH_3_ sensors possess three major bottlenecks: poor selectivity, low concentration detection, and room-temperature operation. State-of-the-art NH_3_ sensors are scaling up using advanced nano-systems possessing rapid, selective, efficient, and enhanced detection to overcome these challenges. MXene–polymer nanocomposites (MXP-NCs) are emerging as advanced nanomaterials of choice for NH_3_ sensing owing to their affordability, excellent conductivity, mechanical flexibility, scalable production, rich surface functionalities, and tunable morphology. The MXP-NCs have demonstrated high performance to develop next-generation intelligent NH_3_ sensors in agricultural, industrial, and biomedical applications. However, their excellent NH_3_-sensing features are not articulated in the form of a review. This comprehensive review summarizes state-of-the-art MXP-NCs fabrication techniques, optimization of desired properties, enhanced sensing characteristics, and applications to detect airborne NH_3_. Furthermore, an overview of challenges, possible solutions, and prospects associated with MXP-NCs is discussed.

## 1. Exploring Needs of Ammonia Detection and Monitoring: Introduction

Ammonia (NH_3_) is extensively used in various commercial products and manufacturing industries, including refrigerants, fertilizers, explosives, hydrogen storage and production, pharmaceuticals, packaging, and chemicals [1,2]. As per statistics, NH_3_ was the second-highest produced chemical commodity, with 235 million tons in 2019 and is predicted to reach around 290 million tons in 2030 [1]. However, NH_3_ is highly toxic to human health and a major environmental contaminant. As per the Occupational Safety and Health Administration (OSHA), the maximum exposure limit of NH_3_ at 25 ppm is 8 h, and when reaching 35 ppm, is 10 min at the workplace [3]. The NH_3_ transforms into ammonium salt on inhalation or absorption through the eyes, nose, skin, or respiratory tract. It can cause illness, lung edema, renal dysfunction, nervous disorder, and mortality exceeding 200–500 mg/kg of body weight [3].

Moreover, NH_3_ is majorly produced from the agricultural sector due to fertilizers at large scale, livestock production, and low nitrogen use efficiency [4]. Airborne NH_3_ possesses a relationship with spatiotemporal disparity of particulate matter (PM 2.5) levels, which is dangerous for public health on penetrating the human body, especially during coronavirus disease (COVID-19) [5,6]. Furthermore, NH_3_ is found in exhaled human breath and acts as a crucial biomarker for diagnosing gastric ulcers and renal disorder caused by *Helicobacter pylori* infection, and recently COVID-19 [7,8]. Renal failure causes the accumulation of urea in the form of ammonium ions in the blood, which penetrate through the lung membrane. As a result, the exhaled breath of patients with peptic ulcer and renal disorder carries NH_3_ in a concentration range of 0.82 to 14.7 ppm [9]. Hence, real-time monitoring of low-level NH_3_ emission has become a priority area for lifesaving, environmental well-being, and biomedical applications. The need of NH_3_ sensing and its state-of-the-art sensing approaches are illustrated in Figure 1.

Various NH_3_ detecting techniques (Figure 2) include calorimetric, gravimetric, electrochemical, optical, chromatographic, laser-based, catalytic, resistive, acoustic, and quartz crystal microbalance-based methods [10,11,12,13,14,15,16,17,18]. Among them, chemiresistors have become critical NH_3_ sensing technologies due to their miniaturized design, low-cost, user-friendly, portable, and energy efficiency to detect NH_3_ (at ~1ppb) in human breath, workplace leak, and agricultural fields [10,13]. Various NH_3_ sensing approaches along with advantages, disadvantages, and potential application are discussed in Table 1.

Chemiresistor is a sensor that consists of a sensing material in the form of a layer deposited over a substrate containing electrodes followed by detecting circuitry [10]. It detects alterations in the conductance of sensing material in the presence of NH_3_. The commercialization of NH_3_ chemiresistors for personalized health, industrial, environmental, and agricultural monitoring is proliferating. NH_3_ monitoring remains an area of active research owing to urgent requirements for reliable, efficient, miniature, portable, and intelligent sensing systems for security and healthcare. The current state-of-the-art in gas sensors for NH_3_ monitoring has already been discussed in literature [2,3,4,5,6,7,8,9,10].

Currently, 2D sensing nanomaterials owing to their larger specific surface area, tunable chemistries, and their fabrication techniques, have been extensively studied in terms of three S’s (sensitivity, selectivity, and stability) and five essential R’s (room temperature operation, range of detection, repeatability, response and recovery, and reproducibility) [11,14,15,16,17,18,19,20,21]. A 2D material (such as graphene and its derivatives, 2D organic frameworks, and polymers, black phosphorus, borophene, metal dichalcogenides, phosphorene, molybdenum disulfide, metal carbides, or nitrides (MXenes)) is generally stated to an atomically thin layer or few layers of flakes with thickness varying from 1–5 nm [14,21]. Although various conventional 2D materials, including graphene or metal dichalcogenides, have shown promising NH_3_ sensing properties due to their high specific surface area and tunable chemistries, they are lacking due to complex synthesis techniques, difficult surface functionalization, and poor selectivity [18,19,20]. MXenes are one of the recent and most popular NH_3_ sensing materials owing to their excellent hydrophilicity, excellent conductivity, large surface to volume ratio, biocompatibility, high mechanical strength, and enriched surface chemistry (due to the presence of surface terminated groups such as –OH, –O or –F) [22,23,24]. Due to excellent metallic type conductivity, MXenes are promising to detect reducing NH_3_ with a higher probability of its oxidation [25]. However, due to high adsorption energy toward NH_3_, they lack complete and rapid recovery [25]_._ MXene-based NH_3_ sensors must be produced for real-world applications in large quantities with high mechanical flexibility for machine processability. Thus, the conductivity and flexibility must be optimized simultaneously, which is still challenging [26,27]. In spite of remarkable features, the use of MXene based sensors for practical applications is somehow limited to the best of our literature knowledge.

Conversely, chemiresistors based on nanostructured polymers (e.g., polypyrrole (PPy), polyvinyl alcohol (PVA), polyaniline (PAN), poly (aniline co-pyrrole) (PAP), polystyrene (PS), poly 3,4-ethylene dioxythiophene (PEDOT), and cationic polyacrylamide (CPAM)) of desired functionality and conductivity continue to enhance NH_3_ sensing performance [10,28,29,30,31]. Polymer-based NH_3_ sensors are popular for miniaturized configuration, user, and environmentally friendly, tunable chemistries, low-cost, flexibility, and energy efficiency. However, owing to the high affinity of polymers toward humidity and volatile organic compounds present in the environment, it exhibits poor stability and cross-sensitivity [10,28,29,30].

Important issues related to cross-sensitivity and poor stability of polymers, and the poor recovery and difficult processability of MXene, etc., hinder their versatile utilizations in NH_3_ sensor fabrication. The use of hybrid nanocomposites of these two materials may result in NH_3_ sensors with efficient and enhanced NH_3_ sensing characteristics. Hybrid nanocomposites are a multi-component material in which at least one of the parent components are of nanoscale dimensions and can be classified as intercalated systems, phase-separated systems, and exfoliated systems [32,33]. It has been predicted that host−guest chemistry and the use of polymer and MXene counterparts in hybrid nanocomposites collectively help to eliminate their shortcomings owing to complementary or synergetic effects, leading to the development of improved NH_3_ sensors [27,34]. MXene–polymer nanocomposites have recently aroused inordinate interest in NH_3_ sensing applications. However, there are reports on nanocomposites of polymers with other 2D materials such as graphene and its derivatives, metal dichalcogenides and molybdenum disulfide [11,14,17,18,19,20]. However, the fabrication and functionalization of such nanocomposites is tedious, which limits their application in ammonia monitoring [11,14,17,18,19,20]. As well, the presence of surface functional groups over MXene makes them more prone to polymers for formation of enhanced hybrids through various multi-interactions such as hydrogen bonding, van der Waals forces, covalent bonding or electrostatic interactions [35,36].

Additionally, MXene–polymer nanocomposites (MXP-NCs) have already been explored for outstanding energy storage applications [35,36,37]. As well, various theoretical studies based on density functional theory predict MXP-NCs to possess better affinity toward ammonia [38,39,40]. This proposes MXP-NCs as a promising sensing material for ammonia monitoring as compared to other 2D materials –polymer nanocomposites. There is significant literature dedicated to MXene for gas detection applications; however, the reports on MXene–polymer nanocomposite-based chemiresistors are scarce [25,27]. To this date, there is no dedicated systematic review on MXene–polymer nanocomposites-based NH_3_ chemiresistors in the literature. With this motivation, this review focuses on MXene–polymer nanocomposites-based NH_3_ sensors and acts as a fundamental work structure to lead future research. This review emphasizes various prospects related to MXP-NCs types, their fabrication and structures, monitoring of NH_3_ in the environment, agricultural fields, and biomedical applications. Additionally, efforts have been made to discuss the NH_3_ sensing mechanism of MXene–polymer nanocomposites. However, better optimization is required for the commercialization of nanocomposite-based NH_3_ sensors.

## 2. Exploring MXP-NCs for Efficient NH_3_ Sensing

The MXP-NCs are rapidly growing in the research of advanced functional materials (Figure 3). MXP-NCs are combinations of nanoscale MXene and polymer counterparts, wherein multi-interactions at the molecular level generate exceptional characteristics at the interface.

Since 2014, the research group of Gogotsi has been exploring the origin, fundamentals, developments, prospects, and potential applications of MXP-NCs [23]. This group has explored MXene and MXene–polymer hybrid composites and anticipated this area of research as a multidisciplinary field. Their first report was on the fabrication of a hybrid nanocomposite of Ti_3_C_2_T_z_, the most widely studied MXene to date, with PVA and PDDA [37]. Although there is extensive research dedicated to MXP-NCs after this first report, the scale is smaller than the studies on MXene itself [38]. Various polymers, including hydrophilic (such as PVA, PS, PA, PI, silicones) and conducting (PAN, PVP, PPy, PEDOT, PSS), have been used for making MXP hybrids [27,38,39,40]. Efforts have been dedicated to studying, optimization and improving of properties generated at the interfaces through molecular and supramolecular dynamics of MXP hybrid nanocomposites [27,38,39,40]. Experimental studies have revealed that incorporation of polymers in the MXene matrix or vice-versa leads to MXP-NCs with enhanced and improved properties [27,38,39,40]. The MXP-NCs offer diverse characteristics absent in their precursors, such as excellent conductivity and flexible mechanical properties due to interfacial multi-interactions among the two phases. Therefore, MXP-NCs have been predicted for various applications such as energy generation and storage, electromagnetic shielding, sensors, actuators, and optical limiters.

A promising and demanding field of research to investigate the applications is based on utilizing MXP-NCs as gas-sensing materials. Many reports based on density functional theory (DFT) have predicted the specific affinity of MXP-NCs toward NH_3_, which has led to dedicated research toward MXP-NCs-based NH_3_ sensors [38,39,40]. These MXP-NCs are suspected to demonstrate exceptional hetero-interfacial effects, enhancing interfacial charge transfer and contact with NH_3_ molecules due to abundant surface functionalities and synergistic effects. Other than interfacial considerations, the effect of various NC parameters such as NC composition and fabrication technique is still a matter of dedicated interest to achieve sensitive and selective detection. Hence, in developing MXP-NCs, a long-term goal is to advance new fabrication strategies to control morphology and composition. This control allows the optimization of MXP-NCs properties (including electrical, mechanical, and optical), which secondarily impacts their NH_3_ sensing performances. The classification of MXP-NCs based on their composition and adopted techniques to synthesize MXP-NCs for gas sensing applications is discussed below.

## 3. Classification of MXP-NCs Functional Structures

As a new family to 2D inorganic materials, MXene is at the infancy stage, obtained from selective elimination of “A” from its precursor MAX phase [23]. The MAX phases are hexagonal layered early transition metal carbides and nitrides with a composition represented by M_n+1_AX_n_ (n = 1, 2 or 3), where “M” is an early transition material, A is 13 or 14 group element, and “X” can be carbon, nitrogen, or both [23]. It has been revealed that there are around 70 MAX phases with ordered double early transition metal structures and more than 30 MXenes. Generally, MXenes are represented by the universal formula M_n+1_X_n_T_x_ with (n + 1) layers of “M” are interleaved with “n” layers of “X”. Various surface functionalities (including hydroxyl, oxygen, fluorine, and chlorine) are represented by “T” in the formula, resulting from different fabrication approaches [23]. Among all, M_3 × 2_T_x_ and M_2_X_2_T_x_ have been evaluated for NH_3_ sensing performance. As revealed by Lee et al. [41], titanium carbide (Ti_3_C_2_T_x_) was the first MXene (M_3_X_2_T_x_) reported for NH_3_ sensing, followed by specific reports [42]. Further, it has been revealed that M_2_X_2_T_x_-MXenes (such as niobium carbide: Nb_2_CT_x_) possess higher specific surface areas owing to their fewer atomic layers compared to M_3_X_2_T_x_-MXenes [43] and are a potential candidate for NH_3_ sensing. However, the practical applications of pristine MXene sensors are inhibited due to few drawbacks such as easy restacking, low flexibility, and poor stability in an oxygen atmosphere [44,45,46].

Taking advantage of easy processability, low cost, mechanical flexibility, and low toxicity of macromolecules, the combination of MXenes and polymers can further enhance the NH_3_ sensing performance of their nanocomposites, including mechanical flexibility, electrical conductivity, and solution solubility. Depending upon the nature and structure of precursor MXene (variation in ‘n’), MXP-NCs can be generally represented by formula M_n+1_X_n_T_x_-P with “P” representing precursor polymer or polymer combination. To date, M_3_X_2_T_x_-P and M_2_X_2_T_x_-P classes of MXP-NCs have only been studied for NH_3_ sensing due to ease of synthesis and stability compared to other MXenes [25,27].

As revealed, polymer addition leads to exfoliation of MXene due to a surge in the interlayer distance. It inhibits the stacking of MXene layers and leads to the formation of hierarchal lamellar structures. Depending upon the nature of the polymer, interfacial multi-interactions (Figure 4) including covalent interaction (poly(2-(dimethylamino)ethyl methacrylate: (PDMAEMA) [47], electrostatic interaction (PAN, PEDOT) [48,49] and hydrogen bonds (PPY, PAM, PSS) [50,51] results in the formation of MXP-NCs with MXene layers stacked by polymer chains. Li et al. [52] reported forming a hierarchy of core-shell-type PAN wrapped Ti_3_C_2_T_x_ nanosheets accompanied by PAN dendrites. The dendritic structure over PAN nanoparticles increases the interlayer distance in lamellar structure, enlarging its specific surface area. Hence, the state-of-the-art fabrication of MXP-NCs is dedicated to developing new fabrication techniques to optimize morphology and properties. The adopted approaches to design M_n+1_X_n_T_x_-P NCs with desired properties are discussed below.

## 4. Fabrication of MXP-NCs-Based High-Performance NH_3_ Sensors

The exceptional properties of MXP-NCs for NH_3_ sensing result from constant advancements in its fabrication technique. These developments are planned to efficiently optimize the required properties of target sensing material such as morphology, topology, crystallinity, and electrical, mechanical, and thermal properties. Several ways to develop state-of-the-art MXP-NCs based NH_3_ sensors generally consist of three stage procedures: synthesis of MXene, synthesis of MXP-NCs, and fabrication of chemiresistor.

### 4.1. Stage-1: Synthesis of MXene

The synthesis of MXene is the primary stage to develop a state-of-the-art MXP-NCs based NH_3_ sensing device [38,39]. Recently, many techniques to synthesize MXene have been developed, which can be briefed into three main strategies: selective etching techniques (Wet etching from precursors, molten-salt etching), bottom-up techniques (chemical vapor deposition: CVD, salt templated growth) and chemical transformations (ammoniation, deoxygenation and carburization). Numerous advancements in these techniques are already extensively reviewed [23,42].

A typical top-down etching-based approach consists of three basic steps: etching, washing, and delamination [23,53] (Figure 5). Etching involves removal of “A” element from MAX phase through the action of etchants such as hydrofluoric acid (HF), lithium fluoride (LiF), hydrochloric acid (HCl), or their combinations. It is followed by washing through centrifugation, decantation, or dispersing into freshwater, in which the pH of a remnant is adjusted to neutral. Lastly, single-layer MXene or few layers are obtained through manual shaking, ultrasonication, or by adding intercalants under delamination process [23,42]. The chemical mechanism associated with the synthesis of MXene using preferential HF etching process can be elaborately demonstrated as follows [23,39,54,55]:

M_n+1_AX_n_ + 3HF → AF_3_ + 3/2 H_2_ + M_n+1_X_n_

M_n+1_X_n_ + 2H_2_O → M_n+1_X_n_(OH)_2_ + H_2_

M_n+1_X_n_ + 2HF → M_n+1_X_n_F_2_ + H_2_

In a typical etching based synthesis, Wang et al. [56,57] used HF etching followed by tetrapropylammonium hydroxide (TPOH) intercalation process to achieve ultrathin 2D niobium carbide MXene nanosheets (Nb_2_CT_x_) from niobium aluminum carbide MAX phase (Nb_2_AlC). A similar procedure was adopted by Li et al. [52] in the synthesis of 2D titanium carbide (Ti_3_C_2_T_x_) nanosheets from titanium aluminum carbide (Ti_3_AlC_2_) using HF etching and TPOH intercalation.

There are other reports on HF-free synthesis of MXenes by substituting it with ammonium fluoride (NH_4_F) aqueous etching method to avoid the toxicity and corrosive nature of HF [23,58]. Xie et al. [59] substituted HF with HCl to fabricate Ti_3_C_2_T_x_, but the obtained MXene was not found to be ideal. Bottom-up approaches include chemical vapor deposition (CVD) and high-temperature molten salt method, but those are complex, time-consuming, and costly techniques [23,39,60]. As well, there are no reports on the synthesis of MXene using a bottom-up approach to develop state-of-the-art MXP-NCs based NH_3_ sensors.

### 4.2. Stage-2: Synthesis of MXP-NCs

This stage is composed of synthesizing MXP-NCs using various strategies. The strategies reported for the synthesis of MXP-NCs can be summarized into two major approaches, including ex situ and in situ routes [38,39,40] (Figure 6). Ex situ routes include blending of separately synthesized precursor materials (MXenes and polymers). In contrast, in situ routes involve either the synthesis of both precursors and synthesizing one precursor in the presence of others. However, other reports on novel routes include RIR-MAPLE deposition, roller mill assisted synthesis and emulsion-based synthesis for stage-2 [27,39] NH_3_ sensor chemiresistor fabrication. Hence, this review will majorly emphasize in situ and ex situ routes.

#### 4.2.1. Ex situ Routes

The ex situ synthesis routes are more commonly applied since it is easy to control the composition and properties of MXP-NCs. Taking advantage of weak interaction (electrostatic interaction, van der Waals attraction, and hydrogen bonding force) between polymer and MXene precursors, MXP-NCs are mainly synthesized by ex situ approaches. In ex situ routes, presynthesized precursors (MXene and polymer) are mixed through various techniques such as blending (solvent processing, wet spinning methods), alternate deposition, and melt processing to design MXP-NCs [27,38,39].

Majorly, blending techniques are applied for hydrophilic precursors, in which presynthesized precursors are blended in a common solvent [25,38,39]. In a typical synthesis, aqueous or colloidal MXene is added to the polymer solution. The solvent is removed by a suitable technique such as vacuum filtration, evaporation, and precipitation [39]. During wet spinning, an aqueous solution of precursors is blended through centrifugation to obtain MXP-NCs. Zhao et al. [61] fabricated Ti_3_C_2_T_x_/CPAM NC by stirring the aqueous solution of Ti_3_C_2_T_x_ and CPAM through centrifugation. However, in solution blending, two different precursor solutions (which are mutually soluble) are used [39]. Naguib et al. [62] reported Ti_3_C_2_T_x_/PAM NC fabrication by blending two distinct dispersions of MXene in dimethylsulfoxide (DMSO) and aqueous PAM. When both the precursors are blended, the polymer chains push the MXene sheets to the interstitial space among the chains, leading to the formation of a 3D continuous hierarchal network of conductive sheets. This continuous linked conductive network and lamellar structure are essential to design a state-of-the-art sensor. Hence, blending routes are advantageous due to the exclusion of toxic solvents, ease of processing, and ability to modulate the physical and chemical properties by controlling the concentration of precursors [38,39]. However, these approaches are limited due to the nature of solvent and polymer. Polar solvents (including protic and aprotic) can only be used in blending approaches due to their intrinsic compatibility with MXenes [38].

Similarly, only soluble or solution-dispersible polymers can be processed using these routes. In contrast, nonpolar or weekly polar polymers require surface modifications to increase solution dispersibility before solvent blending. Yu et al. [63] reported the use of cetyltrimethylammonium bromide (CTAB) to increase the dispersion of Ti_3_C_2_T_x_ in thermoplastic polyurethane (TPU). Recently, Wang et al. [56] reported an alternate layer deposition technique for Nb_2_CT_x_/PAN-based NC-based NH_3_ sensor fabrication. The Nb_2_CT_x_/PAN NC was obtained by sequential deposition of the PAN layer over polyimide surface through in situ polymerization and Nb_2_CT_x_ layer using spray coating. Since neither of the precursors was deposited in the presence of others, the approach belongs to the ex situ route. The route is advantageous to control the thickness and composition of precursors in NC and possess the ease of processing. However, it lacks in the formation of intrinsic hetero-interfacial junctions among NC.

MXene is incorporated into the polymer matrix in melt processing through thermal techniques, including hot pressing, extrusion, and injection molding conducted above the melting point of precursor polymer [38,39]. Sheng et al. [64] reported the fabrication of Ti_3_C_2_T_x_/polyethylene glycol (PEG) NC using the melt-processing technique. This technique is advantageous for the large-scale processing of MXP-NCs. However, this technique can only be used for polymers that can be melt-processed [38]. In addition, high-temperature processing may degrade precursors, which impacts the intrinsic properties of fabricated MXP-NC [39].

Though ex situ routes are majorly used for the fabrication of MXP-NCs, their commercial applications are limited due to various drawbacks such as struggle in evaporation of solvent, poor flatness, lack of the intrinsic linkage between precursors, substantial waste, less mechanical strength, non-uniform inclusion of precursors, and environmental contamination.

#### 4.2.2. In situ Routes

In in situ approaches, both precursors are either synthesized together or one in the presence of another [38]. It has been reported that for the fabrication of MXP-NCs, a specific monomer has been polymerized on presynthesized MXene using in situ polymerization [27]. These approaches can be summarized into two broad categories: solvent-assisted methods (spin coating, dip coating, drop-casting, inkjet printing, spray coating) and heat-assisted ring-opening polymerization [39]. For solvent-based processing, the desired amount of monomer and MXene is mixed in an appropriate solvent and further added with a suitable amount of polymerization agents, including dopant, oxidant, and surfactant, for a definite amount of time. Typically, the synthesis of NC is confirmed by markers indicating completion of polymerization, such as a change in color (pale yellow or green for PAN). There are reports on the fabrication of PAN/MXene by chemically polymerizing aniline via oxidant APS (ammonium persulfate) and HCl (dopant) over Ti_3_C_2_T_x_ nanosheets. Jin et al. [65] reported the fabrication of PEDOT: PSS/Ti_3_C_2_T_x_ NC through in situ polymerization of EDOT on Ti_3_C_2_T_x_ sheets in the presence of PSS. Furthermore, Qin et al. [66] utilized in situ electrodeposition to obtain MXene/PPy nanocomposite with a 3D porous structure. Carey et al. [67] obtained polyamide (PA)-6-based MXene NC utilizing thermally assisted ring-opening polymerization. Typically, a mixture of monomer and MXene is heated at elevated temperatures (above the melting point of polymer) under ambient conditions.

The authors have further demonstrated in their respective reports that in situ routes improve dispersity, electrical, optical, and chemical properties [25,38,39]. The advantage of the in situ route is to obtain well-dispersed precursors, which results in the formation of hetero-interfacial junctions within the NC structure. The unique advantage of this approach is the formation of polymer chains on the MXene surface, which leads to exfoliation and expansion of layers of MXene-layered structures [36,38,39]. It helps in achieving a larger specific surface area with more significant porosity. The formation of hetero-interfacial junctions and increase in specific surface area is essential to design a state-of-the-art sensor. However, only a small amount of monomers can be polymerized over the MXene surface due to insufficient polymerization energy. As revealed by Chen et al. [39], there are two primary types of electron transfer during charge transfer induced polymerization of EDOT, including either shift of electrons from highest occupied molecular orbitals (HOMO) of monomer to lowest unoccupied molecular orbitals (LUMO) of MXene or HOMO of MXene to LUMO of monomers. In addition, the monomer must be close to the MXene layer to complete charge transfer during polymerization. Hence, both the processing routes own merits and demerits in designing state-of-the-art sensors, and advances in them are currently researched and developed.

### 4.3. Stage-3: Fabrication of Chemiresistor

Chemiresistors are processed in the final stage by fabricating a sensing film over a suitable substrate utilizing various techniques (Figure 7) such as drop-casting, electrospinning, dip coating, and inkjet printing [17,25,36,45,68]. Various flexible and nonflexible substrates used in the processing of chemiresistors include a glass substrate, fluorine-doped tin oxide (FTO) substrate, indium-tin-oxide (ITO) substrate, polyindole (PI), and polyethylene terephthalate (PET) [10,11,27,42]. Furthermore, suitable conducting electrode configurations such as parallel and interdigitated configured electrodes are deposited on the sensing layer/substrate utilizing suitable techniques such as thermal evaporation, inkjet printing, mask coating, and dip coating [11]. Typically, electrodes of excellent conductors (such as gold (Au)) are used to avoid the presence of noise due to the Schottky effect [10,17].

There are reports on using a dip-coating approach to obtain MXP-NCs over PI substrates with Au interdigitated electrodes [52,65]. Wang et al. [56,57] fabricated a Nb_2_CT_x_/PAN chemiresistor by spray coating prepared MXP-NCs over PI substrate (with Au interdigitated electrodes) in two different reports. However, Zhao et al. [61] used PET substrate with interdigitated silver electrodes to fabricate the CPAM/Ti_3_C_2_T_x_ based NH_3_ sensor. Few reports on the fabrication of NH_3_ sensors suggest dispersing polymeric systems such as PAP or its composites such as polyaniline–silver nanocomposite in m-cresol on glass substrates [3,69]. However, those are not flexible and are not reported for MXP-NCs to date. Hence, the processing of state-of-the-art NH_3_ sensors is accomplished in these three stages, and the electrical, molecular, mechanical, structural, and thermal properties of MXP-NCs can be optimized using appropriate synthesis routes and varying operational parameters or precursors.

## 5. Unique Properties of MXP-NCs for NH_3_ Sensing

Generally, the key to any application lies in tuning the physical and chemical properties of materials used. For instance, controlling conductivity and morphology of sensing material is a crucial parameter in state-of-the-art sensors. It has been revealed that materials with superior conductivity are utilized to detect reducing analytes such as NH_3_ [11,42,69] and that lower conductivities are suitable for monitoring oxidizing analytes such as sulfur dioxide [70,71,72]. Hence, the desired properties of sensing materials are optimized during the synthesis process by varying operational parameters. Mostly, the critical advantage of NCs is the specificity of their properties since specific materials can be utilized to detect specific analytes selectively. MXP-NCs have been proven to be excellent sensing materials with improved properties such as superior conductivity, porous morphology, improved thermal property, enhanced mechanical strength, and flexibility. In addition, oxidation of MXenes is significantly alleviated due to the casing of MXene through polymer [33,38,40]. Advancements in various properties of MXP-NCs prone to improved NH_3_ sensing are discussed below.

### 5.1. Morphological Properties and Molecular Interactions

Morphology and molecular properties are the most critical parameters to determining the sensing performance of MXP-NCs NH_3_ sensors. The specific surface area, porosity, and roughness of sensing materials are essential for sensing performance since sensing is a surface phenomenon. It has been revealed that the inclusion of polymer exfoliates MXene layers, resulting in higher specific areas and improved porosity [39]. The unique morphologies of NCs with optimum porosity have rendered them attractive as the NH_3_ sensing material.

Li et al. [52] reported the hierarchy of core-shell type PAN wrapped Ti_3_C_2_T_x_ nanosheets accompanied by PAN dendrites in Ti_3_C_2_T_x_/PAN NC (Figure 8). It was revealed that the presence of dendrites on PAN nanoparticles has significantly increased the interlayer distance between MXenes, which enlarges its specific surface area and porosity. Jin et al. [65] observed significantly enhanced interlayer separation between Ti_3_C_2_T_x_ nanosheets by introducing PEDOT: PSS. Zhao et al. [61] observed the formation of a multilayer Ti_3_C_2_T_x_ structure with different layers glued by CPAM. Similar results were obtained for M_2_X_2_T_x_-P NCs by Wang et al. [56], utilizing in situ and ex situ synthesis approaches in distinct reports. As a product of the ex situ route, the formation of porous PAN nanofibers coated by ultrathin Nb_2_CT_x_ nanosheets was observed. In addition, these conductive NC fibers were reported to form heterogeneous interfaces owing to multi-interactions among precursors.

In contrast, 3D porous NC film formation due to the growth of PAN nanofibers over 2D Nb_2_CT_x_ nanosheets was fabricated by an in situ route [57]. The formation of 3D spatial structured NC was ascribed to the multi-interaction phenomenon (hydrogen bonding, electrostatic interactions, and van der Waals forces) due to surface functionalities. Numerous irregular holes over the NC surface were reported to significantly facilitate adsorption and desorption of NH_3_ during sensing phenomena.

Generally, surface functionalities and molecular interactions are explored experimentally by various spectroscopic studies. Wang et al. [57] revealed hydrogen bonding among PAN-Nb_2_CT_x_ NC precursors through the blue shift in Fourier transform infrared spectra of pristine PAN. Zhao et al. [56] revealed the presence of surface defects through Raman spectra with an increase in I(D)/I(G) (the D-peak to G-peak intensity ratio) for Ti_3_C_2_T_x_/CPAM NC compared to that of pristine Ti_3_C_2_T_x_. Similar observations revealing the presence of hydrogen bonding and electrostatic interactions between precursors were observed by Jin et al. [65] in terms of enhanced I(D)/I(G) with a relative shift in D-band and G-band. These interactions and specified surface functionalities are observed to promote NH_3_ adsorption over that of other analyte molecules.

The presence of surface defects with enhanced surface functionalities and multi-interactions significantly promotes the adsorption of analyte molecules. The advancements in NH_3_ sensing materials are concerned with attaining lamellar porous structures with enlarged interlayer distances with significant molecular interactions/defects. These hierarchal layered structures linked by macromolecules promote rapid and extensive adsorption of NH_3_ molecules due to higher specific surface area, large porosity, and enhanced surface functionalities. The current research is dedicated to designing different architectures of MXP-NCs to facilitate molecular interactions.

### 5.2. Electrical Properties of MXP-NCs

Generally, MXenes possess excellent conductivity (around 9880 S/cm) [23,58], and their incorporation in polymer improves the electrical properties of NCs such as electrical conductivity, ionic conductivity, charge transport, and conducting pathways [39]. In addition, the intercalation of polymer into MXenes averts restacking and endorses molecular-level coupling among them. The electrical properties of NCs are generally optimized by tuning the composition of precursors [36,39]. Jin et al. [65] systematically investigated the variation in conductivity of pristine PEDOT: PSS (~0.01 S/cm) with an increase in Ti_3_C_2_T_x_ weight percentage (wt%) in NC. The conductivity first increased (~0.02 S/cm for 8%, 0.02 S/cm for 8%, 0.07 S/cm for 15%) and then decreased (~0.03 for 20% and ~0.02 for 25%) as the addition of Ti_3_C_2_T_x_. The highest conductivity was observed with 15 wt% Ti_3_C_2_T_x_. It can be attributed to the balance between the confinement and percolation threshold of Ti_3_C_2_T_x_, where low Ti_3_C_2_T_x_ addition accelerates charge transport, while the high Ti_3_C_2_T_x_ addition retards it to rigid confinement network. Similar results were obtained from PAN/Nb_2_CT_x_ NCs by Wang et al. [56] by varying the Nb_2_CT_x_ spray volume (0.05 mL to 0.2 mL). They revealed that the larger concentration of Nb_2_CT_x_ (0.2 mL) is not conductive and not suitable for NH_3_ sensing. Hence, the movement of charge carriers is restricted after the percolation threshold of NCs and reveals the importance of optimizing the composition of precursors in NCs.

Li et al. [52] observed the significantly enhanced degree of protonation in PAN due to the addition of Ti_3_C_2_T_x_, which facilitates electron transfer in fabricated NC, resulting in its enhanced sensing characteristics. As revealed from X-ray photoelectron spectra (XPS) (Figure 8), the degree of protonation for PAN/Ti_3_C_2_T_x_ corresponding to the percentage of =NH^+^- and -NH_2_^+^- (58.2%) was higher than that of pristine PAN (41.4%). Similar observations were made for Ti_3_C_2_T_x_/perfluorosulfonic acid (Nafion) composite membrane by Liu et al. [73]. Hence, incorporating MXene into polymer facilitates charge transport by providing better conductive pathways [39,74]. The enhanced charge carrier transport facilitates the potential applications of MXP-NCs in the field of state-of-the-art NH_3_ sensors.

### 5.3. Thermal Properties of MXP-NCs

Generally, thermal properties such as thermal conductivity, thermal expansion, and thermal stability are closely related to material processing and practical applications. The thermal stability of sensing material is the main parameter for its commercial development. According to the first principle DFT calculations, the predicted thermal stability of MXenes is higher than those of most conducting and semiconducting low dimensional materials, indicating their potential to increase the stability of MXP-NCs based sensing devices.

Jin et al. [65] investigated the thermal stability of PEDOT: PSS/Ti_3_C_2_T_x_ NCs by thermogravimetric analysis (TGA). It is revealed that the rate of weight loss with temperature (between room temperature to 300 °C) for NC is slower compared to that of pristine PEDOT: PSS (Figure 9). The enhanced thermal stability of MXP-NC is attributed to multi-electron interactions among its precursors.

Furthermore, many studies [75,76,77] revealed the improvement of thermal conductivity of NC due to the addition of MXenes. It is ascribed to the excellent thermal conductivities of MXenes, which improve the thermal stability of NCs. The increase in thermal conductivity of MXP-NC results in a higher heat dissipation capacity of NC, improving its thermal stability [39]. In addition, it was observed that the incorporation of MXene reduces the random arrangement of a polymer chain, which deteriorates the scattering of phonons [76]. Hence, the enhanced thermal stability of MXP-NCs contributes to extend its sensing application under different temperatures, particularly in harsh environments. However, a direct role of thermal conductivity during NH_3_-sensing phenomena in MXP-NC has not been reported yet.

### 5.4. Other Advanced Properties

In addition, other properties of MXP-NCs have been revealed to be relevant to determining its NH_3_ sensing performances. Mechanical flexibility and mechanical strength are highly desirable for the commercial development of NH_3_ sensor [11,13,78]. MXP-NCs are an example of engineered material with mechanical properties, which possess significant mechanical flexibility due to polymer and enhanced mechanical strength in terms of tensile strength, Young’s modulus, and less crack formation due to the presence of MXene [38]. Zhao et al. [61] reported Ti_3_C_2_T_x_/CPAM NC flexibility for NH_3_ sensing with a bending test. It is revealed that the NCs are strong and flexible due to the binding action of CPAM among Ti_3_C_2_T_x_ layers. It is ascribed to hydrogen bond forces between surface functionalities (amide group) of Ti_3_C_2_T_x_ surface and CPAM chains. However, Li et al. [52] and Wang et al. [57] reported using flexible PI substrates to achieve mechanical flexibility of designed MXP-NC-based NH_3_ sensors. The present reports provide promising strategies for the commercial fabrication of MXP-NCs based sensors with significantly reinforced mechanical properties. However, several unresolved tribological concerns remain due to the distribution of precursors, stress transfer mechanism, size-effects due to precursors, etc.

Recently, it has been revealed that the ordering of molecules/chains in polymer and its composites play a vital role in determining their sensing performances [79]. Generally, polymers are amorphous in nature [3,28,33]. The incorporation of MXene in polymer matrix results in the ordering of its polymer chains up to a certain extent, which turns MXP-NCs into semi-crystalline nature [52,56,61,65]. In addition, Li et al. [52] revealed the reduction in peak intensity and peak width at half-height (FWHM) for PAN peak in Ti_3_C_2_T_x_/PAN NC, which may facilitate charge carrier transport. Chaudhary [79] revealed that the interchain spacing between polymer chains increases with a reduction in FWHM value, which facilitates intrachain hopping of charge carrier and restricts interchain hopping. In amorphous macromolecules, the direct current charge carrier transport is governed by Mott’s variable model, in which charge carriers hop from one site to another in variable range (in 3D) [70,79,80].

However, incorporating foreign material in NCs such as metal or metal oxide may restrict the hop of charge carriers to one dimension, which facilitates the charge carrier transport during sensing phenomena and enhances its sensing performance [79] (Figure 10). However, similar observations have not been reported for MXP-NC to date. Nevertheless, based on the same theory, it can be depicted that the charge carrier transport in polymer precursors of MXP-NCs may be enhanced due to prominent intrachain hopping (owing to reduced FWHM). Hence, MXP-NCs provide better charge transport pathways to conduct charge carriers constituting sensing signals during sensing phenomena. Many efforts have been dedicated to exploring the superior properties of MXP-NCs, which further extend their utility in NH_3_ sensing. However, the precise relationship between physiochemical properties and microstructure of MXP-NCs still requires to be thoroughly investigated, which is essential to guide the architecture of MXP-NCs with outstanding NH_3_ sensing performance.

## 6. Approaches for NH_3_ Gas Sensing

The NH_3_ sensing behavior of MXP-NC sensors is obtained by estimating time-dependent variation in the magnitude of conductance (in terms of resistance, current, or voltage) as a function of NH_3_ concentration at room temperature [53,56,57,61,65]. Typically, an NH_3_ sensing setup consists of a test chamber with provisions for controlling temperature and humidity along with NH_3_ flow (Figure 11). It has been revealed that the exposure to NH_3_ molecules onto an NC-based chemiresistor results in a variation in resistance, which is recorded by the use of a measuring device such as a digital multimeter (DMM) or source meter unit (SMU) interfaced to a computer through software such as LAB View [34,57,69]. Simultaneously, NH_3_ flow is cut off, and nitrogen/air is introduced into the test chamber, which restores the original resistance magnitude. NH_3_ sensing measurements are performed by two strategies, including either static mode or dynamic mode. A definite NH_3_ concentration is introduced into a test chamber during static mode measurements, and time-dependent change in resistance is recorded until saturation [65]. Jin et al. [65] used static mode to record NH_3_ sensing performance of Ti_3_C_2_T_x_/PEDOT: PSS NC, in which NH_3_ is added to an air-tight test chamber through a microsyringe. However, in dynamic mode, a continuous flow of definite NH_3_ concentration is maintained by a mass flow controller (MFC), and the corresponding change in resistance is monitored [57]. Li et al. [52] maintained a constant flow of NH_3_ through the test chamber during dynamic measurement of the time-dependent variation of resistance in Ti_3_C_2_T_x_/PAN NC. These two standard strategies are majorly reported to analyze NH_3_ performance in current state-of-the-art sensors.

## 7. NH_3_ Sensing Mechanism Supported by MXP-NCs

The resistance of MXP-NCs varies time-dependently as a function of NH_3_ concentration. The observed variation in resistance of MXP-NCs in NH_3_ is primarily ascribed to synergistic effect, charge transfer amid NC and NH_3_ molecules, and redox reactions among NC and NH_3_ [52,65]. It has been revealed that the electrical conductivity of MXenes varies from metallic to semiconducting regime depending upon the nature of “M”, “X” and surface functionalities [58,81]. In addition, the incorporation of polymers into MXenes results in the semiconducting behavior of MXP-NCs [38,39]. However, the shift in electrical conducting behavior of MXP-NCs depends on its precursors’ nature and composition. Hence, optimizing surface functionalities and precursor concentration to achieve desired conductivity are current challenges in state-of-the-art sensors.

### 7.1. Ammonia Sensing Mechanism in Pristine Precursors of MXP-NCs

To develop a better understanding of an ammonia-sensing mechanism in MXP-NCs, it is essential to understand the interaction of ammonia with pristine precursor materials. Hence, by unifying the interaction mechanism of ammonia with individual precursors, a theory for ammonia sensing through NC can be developed.

Ti_3_C_2_T_x_ and Nb_2_CT_x_ are MXenes precursors, and p-type conducting polymers (such as PAN, PEDOT, etc.) are polymer precursors majorly utilized to design MXP-NCs-based NH_3_ sensors. Hence, the NH_3_ sensing mechanism exclusively depends on heterojunctions/interfaces between the precursors of MXP-NCs. However, understanding the NH_3_ sensing mechanism of precursors of NCs plays a crucial role in exploring that of NC. The p-type conducting polymers (such as PAN, PPy) undergo protonation/deprotonation on adsorption/desorption of NH_3_ molecules [3,82].

The interaction of p-polymer with NH_3_ results in the compensation of charge carriers by transfer of protons or electrons from polymer ring to NH_3_ molecule [83,84] and can be represented as adsorption–desorption phenomenon given by
$$P^{+}X^{-}+{\mathrm{NH}}_{3}\delta^{-}\begin{array}{c} \mathrm{Adsorption}\\ ⇌\\ \mathrm{Desorption} \end{array}P^{0}+{\mathrm{NH}}_{3}^{+}\delta^{-}X^{-}\;\&\;P^{+}X^{-}+{\mathrm{NH}}_{3}\begin{array}{c} \mathrm{Adsorption}\\ ⇌\\ \mathrm{Desorption} \end{array}\;{\left[P-1H\right]}^{0}+{\mathrm{NH}}_{4}^{+}X^{-}$$
where ${\;X}^{-}$ may be any dopant species such as ${\mathrm{Cl}}^{-},{\mathrm{NO}}_{3}^{-},{\mathrm{ClO}}_{4}^{-}{\mathrm{SO}}_{4}^{-}.$ In such an interaction, the number density of charge carriers decreases, resulting in a decrease of conductivity and an increase of electrical resistance of the polymer.

For instance, a change in the redox state of PAN occurs from emeraldine salt to emeraldine based on exposure to NH_3_ molecules. Hence, on exposure to NH_3_, the resistance of p-type polymers increases due to a reduction in its majority charge carrier (hole) density [3,82]. Moreover, Kim et al. [44] revealed that the surface functionalization transforms metallic Ti_3_C_2_T_x_ into a semiconductor of a narrow bandgap. Li et al. [52] revealed that Ti_3_C_2_T_x_ demonstrates p-type sensing behavior toward reducing analytes such as NH_3_. In contrast, Wang et al. [57] revealed the n-type sensing behavior of Nb_2_CT_x_ sheets toward reducing NH_3_. Hence, the NH_3_ sensing mechanism in MXP-NCs is primarily governed by the nature of heterojunctions/interface (p–p-type or p–n-type) between its precursors and secondarily depends upon chemisorption, hydrogen bonding, and formation of unique hetero-interface functions, which are common in both the types of MXP-NCs. These four significant phenomena together govern the NH_3_ sensing mechanism of MXP-NCs and are discussed below.

### 7.2. Ammonia Sensing Mechanism in MXP-NCs

#### 7.2.1. Chemisorption Based Ammonia Sensing

In ambient conditions, the p-type MXP-NCs layer absorbs oxygen from surroundings and converts them into oxygen anions by trapping electrons from the conduction band of NC [85,86]. Hence, it forms an electron depletion layer of high resistance among p-type polymer and p-MXene. The interaction between oxygen anion on the NC surface and adsorbed NH_3_ molecules results in the release of electrons back to conduction band p-MXP-NC [87]. The space charge region varies with an increase of electrons in the conduction band of NC, which decreases its electrical resistance. The original resistance magnitude is restored by cutting off the NH_3_ supply and introducing fresh air due to the replacement of NH_3_ molecules by oxygen anions. Similar observations were confirmed for Ti_3_C_2_T_x_/PEDOT: PSS NC by Jin et al. [65] (Figure 12), where they summarized entire chemisorption phenomena through chemical reactions, including:

O_2_ + e^−^ → O_2_^−^

4NH_3_ + 5O_2_^−^ → 4NO + 5e^−^ + 6H_2_O

However, this phenomenon is limited at room temperature due to the minute conversion of oxygen into oxygen anions [65]. Hence, the contribution of chemisorption phenomena to the net sensing mechanism is infinitesimal at room temperature.

#### 7.2.2. Formation of Hydrogen Bonds

Various theoretical DFT studies have predicted the formation of a large number of intermolecular hydrogen bonds between precursors of MXP-NCs [38,39,57,87]. It has been revealed that the large number of hydrophilic surface functionalities of MXP-NC is occupied by these intermolecular hydrogen bonds, which decreases the water adsorption sites, and in turn, promotes adsorption of NH_3_ molecules [57]. Hence, the formation of hydrogen bonds on MXP-NCs surface contributes to the net-sensing mechanism by facilitating adsorption of NH_3_ molecules.

#### 7.2.3. Formation of Unique Hetero-Interfacial Functional Groups

Various theoretical and experimental studies [36,39,43] have revealed the formation of unique hetero-interfacial functional groups among precursors (MXene and polymer), which transforms the electric behavior of MXP-NCs into semiconducting range. It has been revealed that the presence of unique hetero-interface functionalities facilitates interfacial charge transfer, which in turn enhances NH_3_ sensing performance of MXP-NCs. Similar observations have been reported for Nb_2_CT_x_/PAN and Nb_2_CT_x_/PAN-TENG-based NH_3_ sensors by Wang et al. [56,57] in two separate reports. Hence, hetero-interface functionalities between precursors contribute to the net NH_3_ sensing mechanism by enhancing the interfacial charge carrier transfer.

#### 7.2.4. Physisorption-Based Gas Sensing

Physisorption is the predominant phenomenon contributing to the net NH_3_ sensing mechanism in MXP-NCs. The resistance of MXP-NCs either decreases or increases on exposure to NH_3_ molecules. It is ascribed to the dominance of either precursor over other during sensing phenomena. Jin et al. [65] revealed the decrease of resistance of Ti_3_C_2_T_x_/PEDOT: PSS NC in the presence of NH_3_ due to increased charge carrier concentration. It is attributed to the enhancement of π–π interaction, specific adsorption area, and the number of majority charge carriers of PEDOT: PSS by addition of Ti_3_C_2_T_x_ in NC [65,88]. There is direct charge carrier transfer between NH_3_ molecules and NC surface, which leads to increased charge carriers, decreasing the resistance of NC [65]. However, the explanation is not supported by evidence from the experimental analysis.

In contrast, Li et al. [52] reported an increase in Ti_3_C_2_T_x_/PAN NC resistance on interaction with NH_3_ molecules. It is ascribed to the predominance of PAN and the formation of Schottky junctions in NC [52]. It is revealed that the p-type PAN has a higher work function compared to that of p-type Ti_3_C_2_T_x_, which results in the formation of Schottky junctions at their interfaces with hole depletion layer lying in PAN. The interaction of electron-donating NH_3_ molecules with the NC surface causes a decrease in the width of the depletion layer, which narrows the conductive pathways for charge carriers in PAN during sensing phenomena. Hence, the modulation in the space charge region of NC on interaction with NH_3_ molecules results in increased NC resistance (Figure 13). Similar observations were made for Ti_3_C_2_T_x_/CPAM-based NH_3_ sensor by Zhao et al. [61]. Hence, in p–p-type semiconducting MXP-NCs, the sensing mechanism is governed by the predominance of either precursor or synergistic effects due to the formation of the interface between precursors.

However, in p–n-type MXP-NCs (Nb_2_CT_x_/PAN), Wang et al. [57] revealed the predominance of p-type PAN over n-type Nb_2_CT_x_ during NH_3_ sensing (Figure 14). It was speculated that there is the formation of the p–n junction at the interface between the precursors, which facilitates NH_3_ sensing performance of NC. It results in the formation of a barrier layer between n-Nb2CTx and p–PAN. The interaction of NC surface with NH_3_ molecules causes a decrease in hole concentration (majority carriers) of PAN, which results in broadening the width of the depletion layer toward the PAN side. The narrowed conducting pathways due to broad depletion width at the PAN side increase the resistance of NC [57]. Hence, the net resistance of NC increases on interaction with NH_3_ molecules and restores its original value on flushing through air molecules [56]. A similar mechanism has already been revealed for various organic-inorganic NCs [20,34,46]. Hence, in MXP-NCs with PAN as a precursor, the predominance of PAN over MXene is observed, which can be ascribed to the unique redox nature of PAN and higher affinity toward NH_3_. However, the predominance of any precursor is also a function of the composition and type of NC. The following section focuses on the application of MXP-NCs to detect NH_3_ for environmental monitoring.

## 8. MXP-NCs-Based NH_3_ Sensing Performance

The NH_3_ sensing potential of MXP-NC sensors in terms of various sensing characteristics, including 3S’s, five essential R’s, flexibility, the effect of humidity and temperature, self-driven capability, and real-time monitoring, is discussed in this section. Advancement in sensing layer material and improvement in NH_3_ sensing characteristics are also critically discussed. Sensor fabrication using MXP-NCs for commercial prospects in agriculture, environmental monitoring, bio-medical fields, and workplace safety has also been highlighted.

### 8.1. Sensitivity, Room Temperature Operation, and Low Detection Limit

Sensitivity/sensing response is the crucial sensing characteristic to evaluate the NH_3_ sensing performance of a sensor. Generally, sensitivity toward NH_3_ is represented in terms of the degree of change/percentage change in magnitude of conductance (in terms of resistance or conductivity or current or voltage) on interaction with NH_3_ molecules [34,65,71] and described in terms of:

Degree change (S) of sensing parameter (R) is given by: S = ΔR/R_o_
where ΔR is the change in resistance of chemiresistor on exposure to NH_3_ (R_a_−R_o_) with R_o_ is the stable value of its sensing parameter in ambient conditions, and Ra is the value of sensing parameter in the presence of NH_3_.

Percentage change in sensing parameter: S(%) = (ΔR/R_o_) × 100.

The magnitude of sensitivity majorly depends on NC properties such as morphology, topology, dimension, the surface to volume ratio, effective surface area, porosity, chemical properties, defects, functionalization, composition, bandgap, conductivity, charge transport, charge carrier pathway of sensing material [10,11,18,20,70,89]. Hence, the research is dedicated to improving these properties in state-of-the-art sensors. As mentioned before, the presence of both the precursors (Polymer and MXene) in MXP-NCs contributes to its improved sensing performance, which predicts MXP-NCs as a potential candidate for NH_3_ monitoring. Various reports on MXP-NCs, including M_3_X_2_T_x_-P and M_2_XT_x_-P types, have revealed the enhanced sensitivity of fabricated NCs compared to their precursors (Table 2). A comparative analysis of MXP-NCs, pristine precursors, and other prominent sensing materials are listed in Table 2.

The authors credited the improved sensing behavior of MXP-NCs compared to other materials owing to their higher specific surface area, large adsorption sites, unique surface functionalities, and optimum porosity.

The lowest detection limit (LDL) is defined as the lowest level of concentration of analyte, which can bring significant recordable change in sensing parameter of sensing material. It is essential to specify the application field of designed sensors. For instance, a sensor detecting NH_3_ at less than 0.8 ppm is suitable to act as a biomarker to detect renal failures/gastric ulcers. Among all reported MXP-NC NH_3_ sensors, Wang et al. [57] observed maximum NH_3_ sensitivity with the lowest detection limit (1.19% for 20 ppb NH_3_) Nb_2_CT_x_/PAN NC sensor, which was further evaluated for human breath analysis. They reported Nb_2_CT_x_/PAN NC sensor fabrication through the in situ route, which is suspected of forming hetero-interfacial junctions between its precursors [34]. Improved sensitivity is ascribed to 3D morphology, enhanced interlayer distance, and high specific surface area of NC, along with the presence of hetero-interfacial junctions. Wang et al. [56] measured NH_3_ (100 ppm) sensitivity of Nb_2_CT_x_/PAN NC (197.20%, 301.31%, 108.36%, and 48.90%) as function of Nb_2_CT_x_ concentration (0.05, 0.1, 0.15, and 0.2 mL, respectively). It is revealed that the NH_3_ sensitivity first increased and then decreased with the addition of Nb_2_CT_x_. The highest was observed for 0.1 mL of Nb_2_CT_x_. It can be ascribed to the balance between the adsorption sites and Nb_2_CT_x_ concentration, where after reaching the threshold, Nb_2_CT_x_ hinders the NH_3_ adsorption sites, resulting in a decrease of sensitivity. After reaching the percolation threshold, the conductivity of MXP-NCs decreases, leading to a reduction in the probability of NC oxidation by NH_3_. Similar observations were made for PEDOT: PSS/Ti_3_C_2_T_x_ NC by Jin et al. [65]. The percolation threshold was observed at 15 wt.% of Ti_3_C_2_T_x_ with the highest conductivity of 0.07 S/cm and NH_3_ sensitivity (36.6%).

Further increasing the wt% of Ti_3_C_2_T_x_ in NC resulted in a decrease in its conductivity, decreasing its sensitivity. It is ascribed to optimal interlayer spacing at 15 wt% of Ti_3_C_2_T_x_, which provides a higher specific surface area with large adsorption sites. Hence, the precursor concentration determines NH_3_ sensitivity and LDL of MXP-NCs by modulating its conductivity, interlayer distance, and the number of adsorption sites.

Room temperature operation is essential for commercial advancements of sensors since no micro-heating assemblages are required for their operation [10,11]. It reduces the cost and complexity of fabrication and energy for operation. Additionally, operating the sensor at room temperature prevents sensing material from degradation, increasing its working lifetime [72]. All MXP-NC sensors summarized in Table 2 work at room temperature owing to their enhanced properties, as previously discussed. Hence, MXP-NCs NH_3_ sensors are sensitive, cost-effective, energy-efficient with simple configuration, and possess a high life with a lower detection limit.

### 8.2. Response-Recovery Times, Repeatability, and Detection Range

Response/recovery times are defined as the time taken for sensing response (elevation curve/descent curve) to reach 90% of the saturated change in the sensing response when the sensor is subjected to a step-change in the original value of the sensing parameter [3,17]. They depend on the degree of adsorption/desorption of NH_3_ molecules over sensing material [71]. A suitable sensor is characterized in short response time, indicating rapid detection and short recovery time, indicating faster recovery to its original state. They are essential for repetitive utilization and confirm sensor long service life.

Jin et al. [65] revealed the dependence of response and recovery time of a Ti_3_C_2_T_x_/PEDOT: PSS NC sensor on Ti_3_C_2_T_x_ concentration in terms of conductivity, adsorption sites, and specific surface area. A comparative analysis of reported response and recovery time for various NH_3_ chemiresistors is listed in Table 3. Among all reported MXP-NCs sensors, the lowest response and recovery time (12–14 s) were observed for Ti_3_C_2_T_x_/CPAM by Zhao et al. [61]. It is ascribed to the choice of sensing parameter (current), which illustrates the importance of detecting signals in state-of-the-art sensors. Generally, MXP-NCs containing PAN as polymer precursors show slow response and recovery despite high sensitivity and the lowest detection limit. It can be ascribed to the affinity of PAN toward NH_3_, owing to its redox properties, which results in slower adsorption/desorption of NH_3_ molecules from the NC surface. However, Wang et al. [56,90,91,92,93,94] revealed that the coupling of triboelectric nanogenerators (TENG) with NC sensors results in improved response and recovery speed.

The repeatability of a sensor is characterized by recording the sensing behavior for many consecutive cycles [17]. It governs the stable sensing performance of a sensor without any influence on environmental conditions. A comparative analysis of repeatability results for various NH_3_ sensors is listed in Table 3. The reported MXP-NCs sensors are observed to be repeatable for many consecutive cycles, showing the stability of their sensing response. For instance, Zhao et al. [61] revealed the stability of the CPAM/Ti_3_C_2_T_x_ NC sensor for ten consecutive cycles of NH_3_ exposure.

Furthermore, a sensor is characterized in terms of linear detection range described by its linear fitting regression value (1 for perfect linear detection range) [11,17]. Linear detection range is a crucial parameter for commercial prospects of the sensor, which helps optimize and control sensing performance. Linearity for a sensor is determined by estimating the linear fit from a plot of sensing response as a function of NH_3_ concentration. Jin et al. [65] reported two linear ranges for a fabricated PEDOT: PSS/Ti_3_C_2_T_x_ sensor, including 10–100 ppm (with R^2^ = 0.957) and 100–100 ppm (with R^2^ = 0.983). Two linear regimes are ascribed to an increase in the interaction of NH_3_ molecules with sensor surface with an increase in NH_3_ concentration. The interaction between NH_3_ molecules and sensor is rapid at high concentrations and becomes moderate at low concentrations. Hence, two linear regimes are generally reported for NH_3_ sensors corresponding to low and high NH_3_ concentrations, respectively. However, the sensing response of highly porous sensing materials with large specific surface areas increases at a moderate rate, which results in a single linear regime throughout the detection range [45,56]. Hence, generally, M_3 × 2_T_x_-P NH_3_ sensors consist of two linear detection ranges, and M_2_XT_x_-P NH_3_ sensors show only a single linear detection range. Consequently, improving response and recovery speed while designing NC with a porous architect is the current challenge in state-of-the-art NH_3_ sensors for their commercial prospects.

### 8.3. Selectivity Demonstrated by MXP-NCs-Based NH_3_ Sensors

Cross-sensitivity is the major challenge in the commercial development of gas sensors. A designed sensor/array of sensors is expected to detect specific gas/to distinguish among various gases for future intelligent devices such as the electric nose. The selectivity of the gas sensor is a crucial characteristic for its practical applications, which is observed by comparing its sensing response toward a particular analyte with other interfering analytes [11]. Alternatively, it is measured by recording the sensing response of the sensor toward a particular analyte in the presence of other interfering gases [10,17,95]. However, the selectivity of MXP-NCs NH_3_ sensors has been reported only through comparative analysis [52,57,61,65]. Various reports on MXP-NC sensors reported the selectivity test of designed sensors by exposing them to a specific concentration of different analytes such as ethanol, acetone, methanol, and sulfur dioxide. A comparative analysis of the selectivity test reported for various MXP-NCs is listed in Table 4.

Various MXP-NC sensors are highly selective toward NH_3_ compared to other interfering analytes (Table 4). It is ascribed to the adsorption energy of various analytes at different surfaces [3,28]. Li et al. [52] reported the sensitivity of Ti_3_C_2_T_x_/PAN toward 10 ppm of NH_3_ to be at least one magnitude larger than that of other interfering gases. It is ascribed to unique characteristics of MXP-NCs such as surface functionalities, larger specific surface area, and exclusive affinity of precursors toward NH_3_. Yu et al. [96] revealed the specific selectivity of Ti_2_C-type MXene by theoretical DFT calculations and ascribed it to high adsorption energy of NH_3_ compared to other analytes. Lee et al. [41] described the enhanced selectivity due to functional groups and defects over the MXene surface, which facilitates its bonding and interaction with NH_3_. Wang et al. [56] also showed similar specific selectivity of a designed Nb_2_CT_x_/PAN sensor toward NH_3_. The fabricated sensor exhibits substantial sensitivity (around 9%) toward SO_2_, which is significant considering the literature [34,70,71,72]. However, the sensing signal for SO_2_ can be easily distinguished from that of NH_3_ since the sensing mechanism for both are different due to the opposite nature of gases (NH_3_: reducing, SO_2_: oxidizing) [3,70,71]. Hence, in addition to highly selective MXP-NCs NH_3_ sensor, these reports also open a prospect for detecting SO_2_ through MXP-NCs.

### 8.4. Mechanical Flexibility Demonstrated by MXP-NCs-Based NH_3_ Sensors

Mechanical flexibility is one of the essential characteristics of NH_3_ sensors for its commercial development, which makes them portable, foldable, bendable, and stable in every wear and tear situation [11,13,17]. Mechanically flexible sensors are easy to install at every emission site for monitoring NH_3_ emission/leakage. Generally, flexible sensors are built by utilizing two routes: either by designing self-standing flexible films or using flexible polymer substrates such as PI or PET [13]. The flexibility of NH_3_ sensors is tested by comparing its sensitivity under different bending angles and multiple folds to that of under unbent condition [11,13]. Zhao et al. [61] reported that incorporating CPAM into Ti_3_C_2_T_x_ results in mechanical flexibility of Ti_3_C_2_T_x_/CPAM NC sensing film, which is missing in pristine Ti_3_C_2_T_x_ films. It is ascribed to the gluing action of CPAM, which bonds together Ti_3_C_2_T_x_ stacks, providing its mechanical flexibility. It is revealed that the sensitivity of a pristine Ti_3_C_2_T_x_ sensor varies under flexibility test. However, the sensitivity of the Ti_3_C_2_T_x_/CPAM NC sensor is found stable with negligible descent. There is a slight increase in response on recovery time by approximately 2 s, but that is not significant compared to other sensors. Jin et al. [65] revealed the consistency of a PEDOT: PSS/Ti_3_C_2_T_x_ NCs NH_3_ sensor toward 100 ppm of NH_3_ under various bending angles (Figure 15). Various MXP-NCs-based NH_3_ sensors have shown stable sensitivity under the flexibility test, as illustrated in Table 5.

It has been observed that there are flexibility test reports on M_3 × 2_T_x_-P NH_3_ sensors; however, M_2_XT_x_-P NH_3_ sensors have not been evaluated to date. Hence, MXP-NC NH_3_ sensors are easily processable to attain flexibility due to other polymer precursors or flexible substrates. However, the effects of type of substrate over NH_3_ sensing performance are still unexplored and possess vast scope to state-of-the-art sensors.

### 8.5. Stability-Based on the Effect of Varying Environmental Conditions

Stability is the most prominent factor in determining sensor performance in lifetime and consistency [11]. Stability is majorly affected due to degradation of sensing material on interaction with the analyte and variable surroundings [10]. Under ambient conditions, stability is measured by recording sensing response as a function of time (in days, weeks, or months) [34]. It is crucial to test the stability of MXP-NCs due to the possibility of degradation of its precursors. It has been revealed that polymers degrade with time due to interactions with surroundings, and MXene is prone to oxidation owing to its high reactivity [11,23,29]. However, stability is achieved in organic-inorganic NCs (including MXP-NCs) due to the formation of hetero-interfacial junctions or multi-interactions between its precursors [25,36,38]. Jin et al. [65] reported the stable sensing response (around 33%) of Ti_3_C_2_T_x_/PEDOT: PSS sensor toward 100 ppm of NH_3_ for four weeks. Li et al. [52] and Wang et al. [57] also reported stability of a fabricated MXP-NC sensor for 35 consecutive days with stable sensing response toward 10 ppm of NH_3_. Hence, the MXP-NCs are reported stable in ambient conditions.

However, a sensor must also be stable for commercial purposes in varying temperature and humidity conditions, especially in harsh surroundings [11]. It has already been revealed that polymers (such as PAN) are prone to humidity due to bonding or swelling, which degrades their structure and sensing performance [10,97,98]. Abdulla et al. [99] revealed that the adsorption of water molecules over the PAN surface causes the release of electrons, which results in the formation of H3O^+^ ions and increases its ionic conductivity [100]. The interaction with NH_3_ molecules results in NH_4_OH or NH_3_∙H_2_O formation, which captures protons from PAN, resulting in increased sensitivity [99]. However, after a certain level of humidity, water molecules hinder the adsorption of the NH_3_ molecule limiting its sensitivity [52]. Consequently, there exists a dynamic equilibrium between humidity value and NH_3_ sensing response. Hence, the humidity test is performed by two routes: monitoring sensing response in the presence of NH_3_ [52] and monitoring change in sensing parameters in the absence of NH_3_ [3], with change in humidity.

Jin et al. [65] reported negligible sensing response toward humidity of Ti_3_C_2_T_x_/PEDOT: PSS sensor. However, the NH_3_ sensitivity increases with relative humidity (RH: 20–90%). It is ascribed to increased charge transfer interactions due to a decrease in average separation between water molecules. However, Li et al. [52] revealed that the NH_3_ sensitivity first increases (0–40%RH) and then decreases (40–90%RH) with the increase in RH, and the threshold is found around 40%. The threshold is ascribed to the accumulation of a thin layer of water molecules on the sensor surface, hindering the NH_3_ adsorption sites. Li [52] and Jin et al. [65] used the same MXene precursor (Ti_3_C_2_T_x_) to fabricate MXP-NCs; however, the sensitivity variation with RH is different. It is ascribed to the predominance of polymer precursor (PAN) in Ti_3_C_2_T_x_/PAN NC reported by Li et al. [52], which possess a specific affinity toward NH_3_ [101].

In contrast, Wang et al. [56,57] revealed the threshold of RH for NH_3_ monitoring at 72.2% RH (Figure 16). The interaction of NH_3_ with water molecules present on the sensor surface results in an increase of NH_3_ sensitivity in the range of 41.0–72.2% RH; however, over 72.2% RH sensitivity decreases. It is ascribed to a higher specific surface area of Nb_2_CT_x_ than Ti_3_C_2_T_x_, shifting the threshold RH value to a higher magnitude. It reveals that the threshold RH value of the NH_3_ sensor depends on the nature and type of precursors (Polymer and MXenes). Hence, the influence of humidity on NH_3_ sensing performance is optimized by controlling the nature and composition of precursors in NCs.

Moreover, the study of variation of NH_3_ sensing response as a function of temperature is also vital to evaluate the performance of a sensor and record following the same strategy [11]. It is revealed that the increase in operational temperature results in a linear decrease of NH_3_ sensitivity of MXP-NCs [52,65,94]. It is ascribed to desorption of H_3_O^+^ ions from NC surface and the competition among desorption and exothermic adsorption of NH_3_ on sensor surface at higher temperature [102,103]. Additionally, the increase in the operational temperature of the sensor results in the formation of many oxygen radicals, which causes chemisorption dominance in contributing to the net sensing phenomenon. Similar trends are observed for MXP-NCs sensors by Li et al. [52] and Wang et al. [57], observing a linear decrease in NH_3_ with an increase in operational temperature.

Hence, MXP-NCs-based NH_3_ sensors operate with high stability in ambient and varying surroundings (humidity and varying temperature); however, the optimization of precursors is primarily required to record utmost sensitivity and stability.

## 9. Advancements in NH_3_ Detection for Applications Point-of-View

The inclusion of various technological developments is helping to extensively develop the application prospects of NH_3_ sensors with advanced sensing features. The current need for technological advancements has gained extensive interest in designing intelligent sensors, replacing the current complex, expensive and time-consuming techniques. For instance, monitoring NH_3_ in the agricultural field has advanced with intelligent sensors, reducing the requirement of human resources and efforts. This section highlights the advanced application of next-generation NH_3_ sensors using MXP-NCs as the sensing material.

### 9.1. Human Breath Analysis Based on NH_3_ Detection

The detection of urea/ammonium salt concentration is highly desirable for gastric ulcer/renal patients. It is executed through various complex techniques such as blood test, which are time-consuming, expensive, and requires dedicated human resources. Alternatively, the urea/ammonium salt concentration is detected by monitoring NH_3_ in exhaled human breath [104]. It has been revealed that the NH_3_ concentration in the breath of a renal patient is beyond 0.8 ppm, which can be potentially detected by an NH_3_ sensor with LDL less than 0.8 ppm [104,105]. MXP-NCs sensors are promising biomarkers in detecting NH_3_ in renal/gastric ulcer patients owing to their low detection limit (20 ppb: Table 1) with high sensitivity. Hence, airborne NH_3_ sensors have the potential to present an alternative to present-day monitoring techniques.

Wang et al. [57] revealed the use of an Nb_2_CT_x_/PAN sensor to detect NH_3_ in a human breath through an exhaled-breath analysis simulation system (Figure 17). The sensitivity of the Nb_2_CT_x_/PAN sensor is observed around 10% on exposure to the simulated breath of a patient containing 0.8 ppm of NH_3_. It is ascribed to the fascinating features of MXP-NCs toward NH_3_ detection such as higher specific surface area, large adsorption sites, specific surface functionalities, explicit affinity, and fast conducting pathways due to the presence of both the precursors. However, a study on other sensing characteristics for detailed breath NH_3_ analysis is missing and possesses scope for further advancements. The sensing analysis is required to associate with pattern recognition, array-sensing, and data from clinical trials for its commercial development in bio-medical applications.

### 9.2. Detecting Volatilization of Agricultural NH_3_

Reducing NH_3_ emission by employing advanced farm practices is a priority of environmental regulators in the agricultural field around the world. It has been revealed that the 90% of airborne NH_3_ is released from various agricultural sources such as livestock, fertilizers, etc. [106,107]. Hence, to sustain the implementation of NH_3_ emission mitigation practices, advanced sensors are developed to identify emission sources, individual contributions and assess the effectiveness of control measures. The MXP-NCs owing to their excellent NH_3_ detecting capabilities are promising to detect the volatilization of NH_3_ in the agricultural sector.

Li et al. [52] predicted the feasibility of a Ti_3_C_2_T_x_/PAN sensor to detect volatilized NH_3_ in agricultural applications by designing an experiment based on an agricultural simulator. The NH_3_ volatilization results of the Ti_3_C_2_T_x_/PAN sensor are compared with that of the conventional sulfuric acid adsorption method (SAAM) and Drager nitrogen tube method (DTM). Typically, a layer of fresh soil sprayed with aqueous urea solution is cultivated under dark and ambient conditions in a large vessel. A beaker containing sulfuric acid is placed in the vessel after fertilization, and the concentration of ammonium salt produced due to reaction among acid vapors and urea is recorded (SAAM). Additionally, a DTM tube is placed inside the vessel, which changes color on reacting with volatilized NH_3_ molecules. The color scale of the DTM tube gives a measurement of the concentration of volatilized NH_3_.

Moreover, a Ti_3_C_2_T_x_/PAN sensor is also placed in the same vessel, and monitoring of volatilized NH_3_ is performed. The results obtained using the Ti_3_C_2_T_x_/PAN sensor in predicting NH_3_ trends are similar to those obtained from conventional SAAM and DTM techniques [52] (Figure 18). The study proposes the potential of MXP-NCs sensor in detecting NH_3_ volatilization in agricultural sectors as an intelligent alternative for easy and precise detection with the reduced requirement of human resources. Hence, the MXP-NCs sensors can act as intelligent sensors that give rise to advancements in intelligence for agricultural applications. However, it requires dedicated research interest with advanced intelligent techniques to configure smart sensors with optimized performance.

### 9.3. Self-Driven NH_3_ Sensing

The energy crisis is another primary global concern, resulting in the advancement of self-drive smart sensors eliminating the requirement of conventional power supplies. Self-driven NH_3_ sensors operate through mechanical forces generated by nanogenerators such as TENG instead of conventional electric sources [94,108]. TENG operates on mechanical forces and acts as a voltage source for NH_3_ sensors [109,110]. Wang et al. [56] reported on a Nb_2_CT_x_/PAN–TENG sensor to detect a low trace of NH_3_. The fabricated Nb_2_CT_x_/PAN sensor (on the interdigitated PI substrate) is coupled to a TENG operated through a linear motor of a fixed frequency of 1 Hz (Figure 19A). The coupled TENG acts as a power source to drive the MXP-NC sensor for NH_3_ monitoring operation. The interaction between NH_3_ molecules and the sensor causes a change in real-time resistance of the NC sensor, which further varies the output voltage of externally loaded TENG.

In spite of recording real-time resistance of the sensor, variation in output voltage is measured as a sensing parameter (Figure 19B). The sensitivity of the TENG-coupled Nb_2_CT_x_/PAN sensor is many-fold higher than that of the pristine Nb_2_CT_x_/PAN sensor. It is revealed that the sensitivity of the TENG coupled Nb_2_CT_x_/PAN sensor is 2.57% per ppm of NH_3_ with a low response time of 105 s. It is ascribed to the various factors such as synergistic effects due to the formation of the p–n junction between the precursors, TENG presence, and choice of sensing parameter and measuring technique [56].

Hence, the MXP-NCs sensors perform better while coupled with TENG and possess potential application in smart self-driven NH_3_ sensors with fewer energy requirements. However, the design of compact and optimal TENG with advanced sensing configuration is required to address its commercial prospects.

### 9.4. Monitoring of Environmental Contaminated by Atmospheric NH_3_

Atmospheric NH_3_ is a critical environmental pollutant contributing to acidification of the ecosystem, secondary particles formulation, and eutrophication [111,112]. It plays a primary role in the formation of secondary particulate matter by reacting with airborne acidic species (such as oxides of sulfur and nitrogen) to form ammonium ion-based aerosols, which is the foremost constituent of PM2.5 [112,113]. Ammonium-particulate species contribute to degradation in air quality and visibility and affect atmospheric radiative balance [114]. The PM also acts as a potential carrier of coronavirus through aerosol, conveying the virus to a more considerable distance with increased speed [6,115]. Moreover, the inhalation of ammonium PM can induce damage to respiratory, nervous, and renal systems [7,116]. Hence, the monitoring of airborne ammonium contaminants is the primary concern of environmental stakeholders, especially in a pandemic. MXP-NCs sensors are revealed to detect a low concentration of NH_3_ (as low as 20 ppb) with advanced sensing characteristics such as flexibility, portable, and high selectivity (Table 2, Table 3 and Table 4). Wang et al. [57] revealed high-sensing NH_3_ performance of a flexible Nb_2_CT_x_/PAN sensor (with LDL 20 ppb) at room temperature. The compact size and mechanical flexibility of the designed sensor allow the sensor to be installed at every NH_3_ emission site, making it a potential candidate to monitor atmospheric NH_3_ contamination. Hence, MXP-NCs sensors have enormous latency to replace present-day temperature-assisted environmental NH_3_ sensors.

## 10. Challenges and Alternative Approaches

The applications of MXP-NCs for NH_3_ monitoring require imperative advancements related to precursor processing science and sensor fabrication. The main challenge is to develop moderate and safe synthesis approaches to prepare MXenes with low cost and high yield from the aspect of precursor processing [23,24,25,54,117,118,119]. Although HF-free synthesis routes are proven to be better than HF etching routes with fewer defects on the MXene surface, the presence of cations or extra water between the MXene layers may influence its physiochemical properties. The unstable nature of MXenes such as Ti_3_C_2_T_x_ due to prone to oxidation over long-term operations remains another challenge to address.

### 10.1. Optimization of Concentration of Precursors

A challenge in the fabrication of MXP-NCs is simultaneous optimization of mechanical flexibility with maintaining high conductivity. It is essential to maintain a significant concentration of MXene in MXP-NCs to maintain high electroconductivity. However, the increase in MXene concentration results in a decrease in mechanical stretchability and flexibility. Numerous tribological characterization tests must be performed to evaluate the threshold MXene concentration for each polymer precursor. These facts argue the presence of trade-offs between the NC properties and the precursor concentration for an NC sensor.

Finding the optimal concentration of MXene in NC for fabricating a sensor is challenging. A suitable concentration of MXene is required to form conductive pathways/channels in the polymer matrix. This conductive network can break in a particular strain range due to an increase in interlayer distance between MXene sheets. As the strain increases, the MXene sheets forming a conductive network moves away from each other. Further strain above the threshold will eventually result in the complete breaking of the conductive network. These variations in conductive networks arrangement are manifested by a steady decrease in the transient electrical current through the sensor and the extent of strain experienced by the sensor. However, the high concentration of MXene in MXP-NCs results in the formation of a continuous conductive network throughout the NC in an anticipated strain range. It results in no detectable change in conductance, including at high strains. Hence, the concentration of MXene precursor in NC highly determines the sensitivity of the sensor.

The surface functionalization of MXenes can address these challenges. In various MXP-NCs, surface functionalization is required to enhance the affinity between the precursors. Hydroxyl groups on the MXene surface are reactive and can endure various surface functionalization reactions. The formation of hydrogen bonds between hydroxyl groups on the MXene surface and polymer precursor improves lifespan and workability. It also reduces the percolation threshold of MXene precursor in the NC sensor due to forming a well-oriented ordered MXene network. It has been revealed by Guo et al. [120] that the percolation of the threshold of MXene in a nonreactive polymer is reduced from 40 to 6 wt% due to the formation of hydrogen bonds on surface modifications. These hydrogen bonds also improve the mechanical flexibility and stretchability of the sensor, as they work as stress transfer networks between the precursors. Similar observations were made by Zhang et al. [121] for MXene/polydimethylsiloxane sensor. Hence, surface modification of MXene precursor serves as an alternative approach for simultaneous optimization of electroconductivity and mechanical flexibility.

Another alternative approach is using secondary nanoparticles for surface modification of MXenes in NCs, which results in the formation of mixed-dimensional structures. The secondary nanoparticles can be zero-dimensional (size less than 100 nm: silver quantum dots), one-dimensional (such as carbon nanotubes), two-dimensional (graphene), and three-dimensional (modified graphene oxide). Generally, secondary nanoparticles are modified using coupling agents such as silane to provide them with a positive charge at the surface. It results in electrostatic interaction between MXene and secondary nanoparticles, causing the formation of a ternary NC.

Zhou et al. [122] revealed the improved carbon dioxide sensing performance of ternary nanocomposite of nitrogen-doped MXene, Ti_3_C_2_T_x_ (N-MXene), polyethyleneimine (PEI), and reduced graphene oxide (rGO). It is ascribed to the surface modification of MXene through nitrogen doping and the presence of secondary nanoparticles (rGO). Hence, the sensing performance of MXP-NCs can be improved by surface modification of an MXene precursor through any coupling agent or secondary nanoparticles due to hydrogen bonding/electrostatic interactions.

### 10.2. Slower Response

Additionally, MXP-NCs exhibit a slower response toward NH_3_, limiting its commercial prospects. It can be decreased using a hierarchically interlocked structure (such as Ti_3_C_2_T_x_/CPAM) to fabricate an NH_3_ sensor. Based on reported studies, two strategies can be adopted to design state-of-the-art MXP-NC sensors. First, the form of well-distributed conductive channels of MXene in a polymer matrix with increased interlayer separation in NC. Second, improve the deformability of the sensing layer by improving contact between precursors, resulting in hetero-interfacial junctions.

### 10.3. Mass Production

The technical challenges related to mass production and process integration of MXP-NC sensors should be addressed to meet industrial development. It can be achieved through developing one-pot synthesis approaches, in which MXene flakes can be exfoliated during the in situ polymerization of monomers, aiding the scalable production of MXP-NCs.

Addressing the challenges with alternative technologies will further accelerate novel MXP-NCs sensors’ development and explore its potential in widespread application fields.

## 11. Conclusions, Prospects, and Viewpoints

The last several years have witnessed rapid advancement in the investigation of MXP-NCs for sensing applications due to synergies between precursors’ advantages (polymers and MXene). Generally, the stability and mechanical flexibility of MXene are improved, and the electrical, mechanical, and thermal properties of the polymer are enhanced in MXP-NCs, while other properties are still under examination. This review has focused on the advancements in fabrication strategies, fascinating structures, and unique properties of MXP-NCs NH_3_ sensors. The reported NH_3_ sensing performances of MXP-NC sensors have been comparatively analyzed in several comprehensive tables. In addition, diversified applications of MXP-NCs NH_3_ sensors in medical, industrial, environmental, and agricultural industries have been summarized. As per published literature, MXP-NCs have great potential in NH_3_ monitoring, and their overall sensing performances are better than those of other sensing materials. Nevertheless, the development of MXP-NCs NH_3_ sensors is in its infancy, and there are still many scientific quests remaining before revealing the commercial applications of these materials.

MXP-NCs sensors open a prospect for NH_3_-monitoring applications in diversified fields, including bio-medical, industrial, and agricultural. Contrary to conventional metal oxide-based sensors, their operation at room temperature makes them cost effective and energy efficient due to the exclusion of micro-heating components. These NH_3_ sensors are user friendly due to the exclusion of any toxic heavy metal and contribute to negligible toxic nano-waste. Since they can be operated in chemiresistive mode, they are portable, compact, easily handled, simple to configure, and can be produced on a large scale, dissimilar to spectroscopic or electrochemical sensors. However, studies on optimizing large electrical conductivity for mass production of MXene still require ordinate attention. Their hydrophilic nature makes it easier to fabricate flexible and compact NH_3_ sensors, which can be utilized at every emission site. MXP-NCs NH_3_ sensors also possess the potential to fabricate smart and intelligent technologies to detect NH_3_ volatilization in agricultural fields. The high sensitivity makes MXP-NC sensors rapidly detect minute NH_3_ leakage for workplace safety. However, the optimization of response and recovery time still requires improvement. MXPs-NC NH_3_ sensors can substitute expensive and sophisticated clinical tests to detect urea concentration in gastric or renal patients owing to their low detection limit. They can detect the low concentration of NH_3_ (as low as 20 ppb) in exhaled human breath and can result in an optimized real-time NH_3_ monitoring system with low cost and simple configuration. However, more emphasis on clinical trial data is required to explore its exact potential to replace other urea detecting methods and tests. Hence, MXP-NC sensors give new prospects to NH_3_ monitoring in rapid and low-level detection, including intelligent technologies, environment, and user-friendly, compact, flexible, and portable nature, and high sensitivity.

The advanced NH_3_ sensing features of MXP-NCs are ascribed to their unique structural and functional properties such as sizeable interlayer distance, exfoliation, surface functionalities, formation of hetero-interfacial junctions between the precursors, synergistic effects, enhanced porosity, improved charge transport, high thermal and mechanical stability, and high specific surface area. However, the reports on MXP-NCs-based NH_3_ sensors are scarce and require dedicated attention from the scientific community.

Additionally, polymers are abundant in nature, which has been used for NH_3_ monitoring. The use of advanced and functional polymers can further facilitate the NH_3_ detection characteristics of MXP-NCs. Hence, various combinations of MXene and polymers are yet to fabricate and utilized for NH_3_ sensing.

Moreover, there is a vast scope of research in optimizing the properties of MXP-NCs for NH_3_ sensing through optimizing the nature and concentration of precursors. The NH_3_ sensing mechanism can be experimentally explored using advanced spectroscopic/microscopy systems, including in situ FTIR or in situ Raman or kelvin probe force microscopy (KPFM) techniques in the presence of NH_3_.

Consequently, challenges in the processing of MXP-NCs NH_3_ sensor can be summarized as a requirement of large-scale production without compromising the electrical conductivity, optimizing the concentration of precursors (MXene and polymer) for better sensing characteristics, controlling synergistic effects, optimizing humidity with sensing performance, managing durability along with flexibility, choice of precursors and surface modifications. Hence, the colossal possibilities in NH_3_ sensing are addressing these challenges for the development of novel MXP-NC sensors. Less explored fields such as food engineering and electric nose can also benefit from MXP-NC sensors by incorporating recent advancements such as machine learning and pattern recognition. In the future, MXP-NC sensors (Figure 20) application fields are expected to expand and evolve using artificial intelligence.

## Figures and Tables

**Figure 1 nanomaterials-11-02496-f001:**
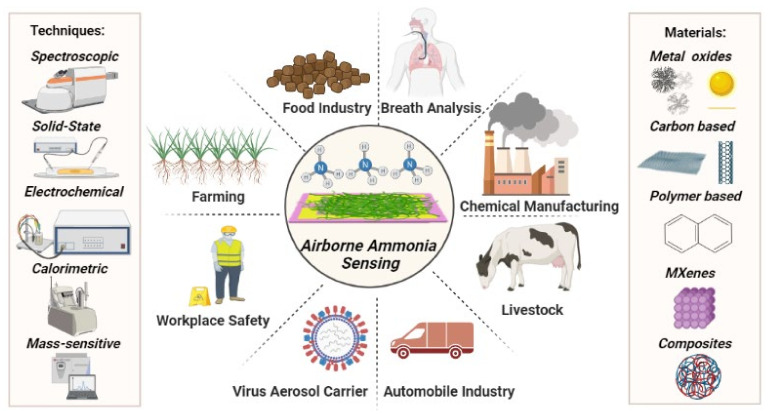
Schematic of state-of-the-art NH_3_ sensing techniques using different sensing signals.

**Figure 2 nanomaterials-11-02496-f002:**
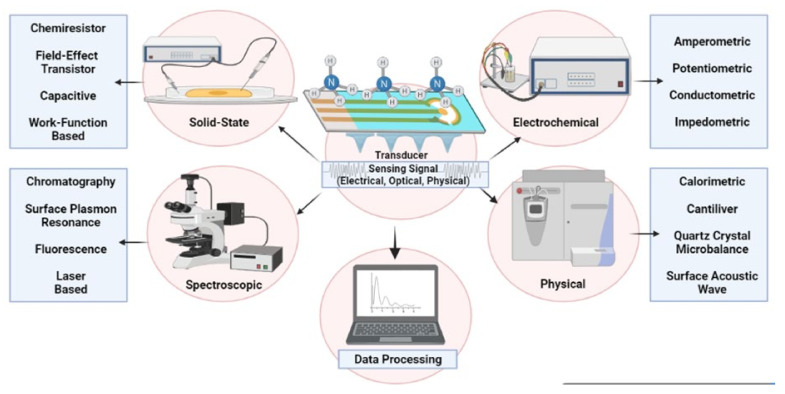
Schematic of various ammonia sensing techniques using different sensing signals.

**Figure 3 nanomaterials-11-02496-f003:**
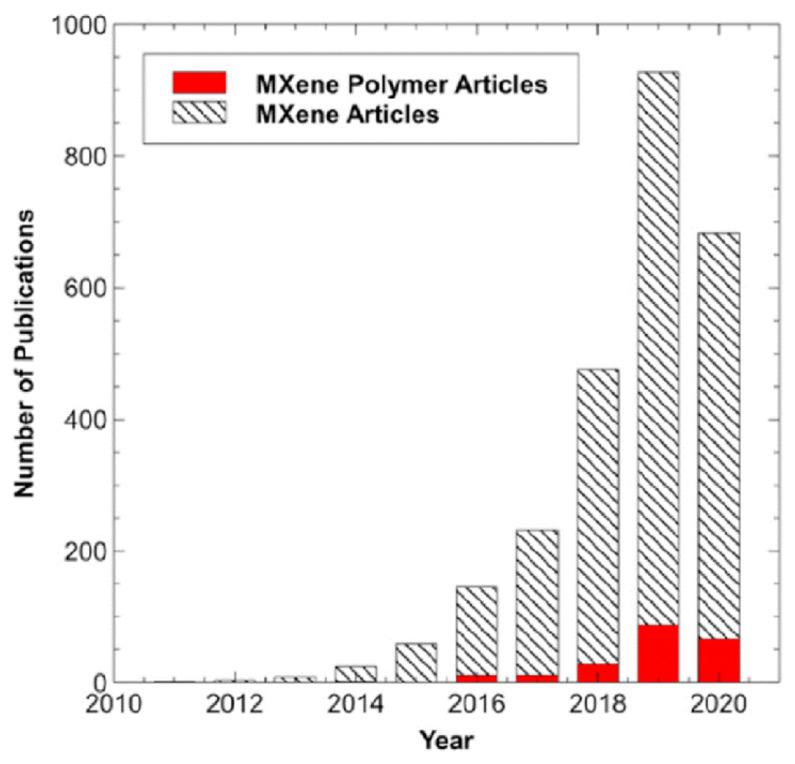
Statistics on yearly publication in the field of MXene–polymer nanocomposites (especially properties and processing of NCs). Reprinted from ref. [38].

**Figure 4 nanomaterials-11-02496-f004:**
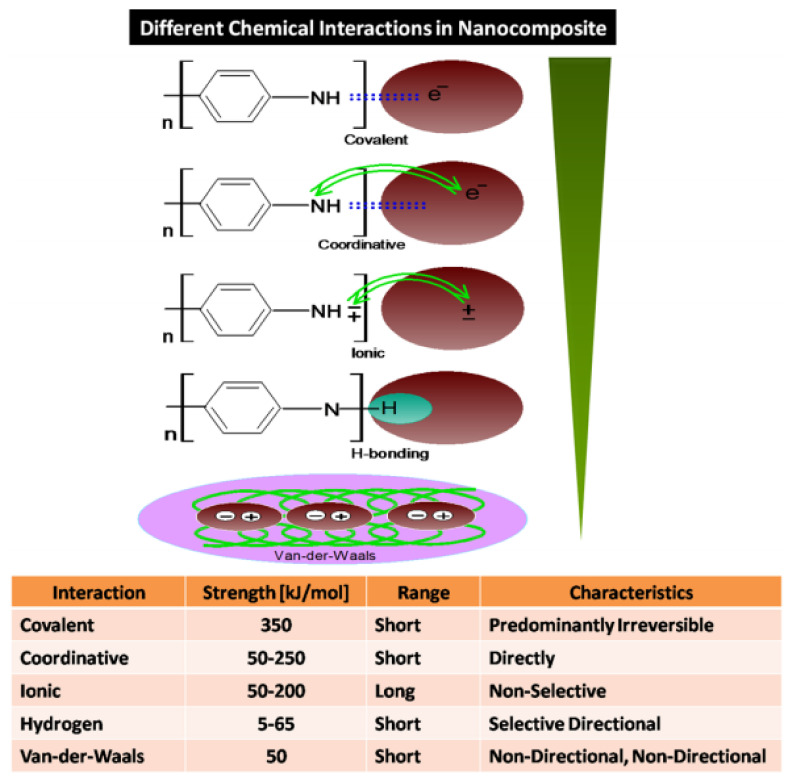
Possible interactions in polymer nanocomposites. Reprinted with permission from ref. [46]. Copyright 2015 American Chemical Society.

**Figure 5 nanomaterials-11-02496-f005:**
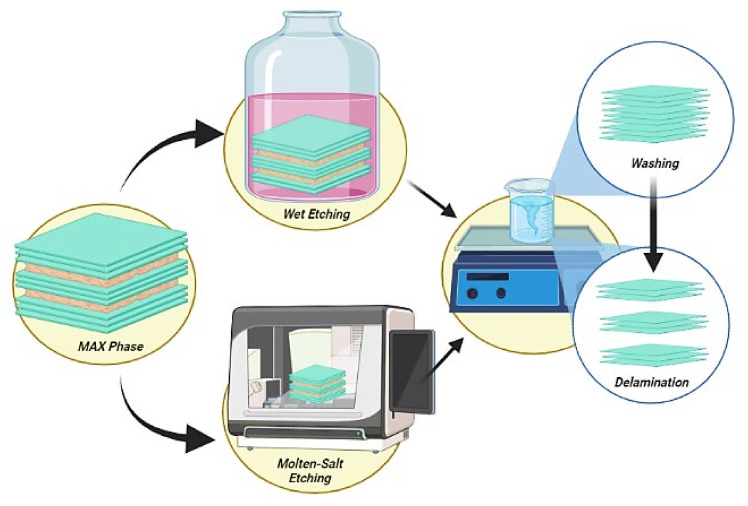
Schematic of fabrication of MXene using selective Etching of MAX phase through wet and dry route.

**Figure 6 nanomaterials-11-02496-f006:**
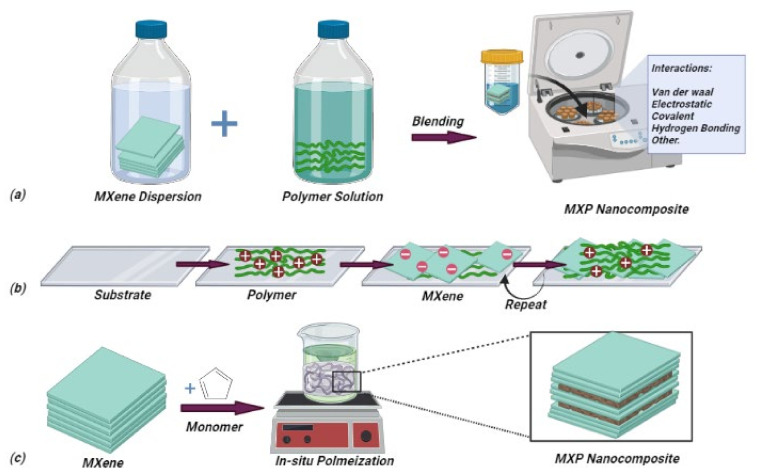
Schematic of synthesis routes for MXP-NCs. Ex situ route: (**a**) blending, (**b**) alternate deposition. In situ route: (**c**) in situ polymerization.

**Figure 7 nanomaterials-11-02496-f007:**
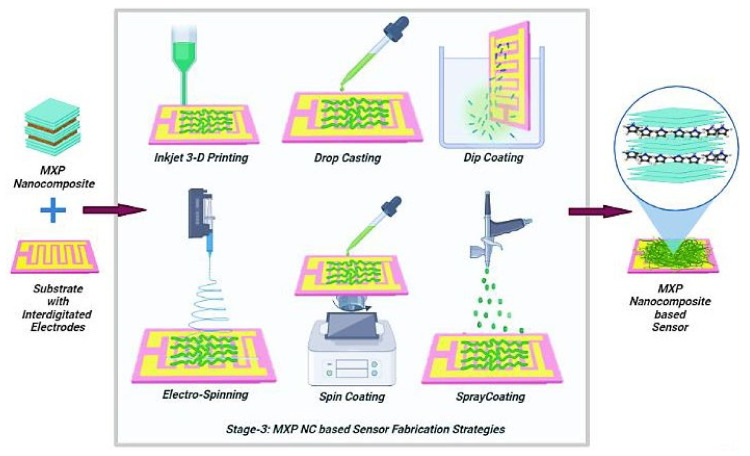
Different reported strategies for fabrication of an ammonia-sensing device using MXP-NCs.

**Figure 8 nanomaterials-11-02496-f008:**
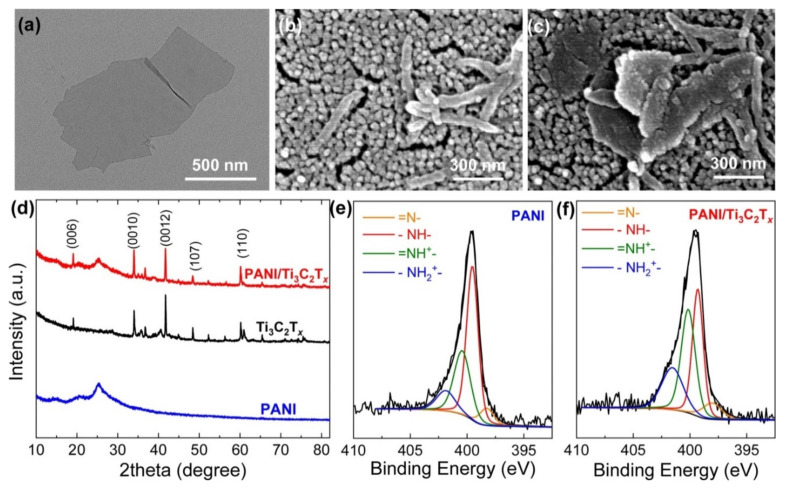
Morphological, structural, and chemical element analysis of PAN/Ti_3_C_2_T_x_ NC with (**a**) Morphology of Pristine Ti_3_C_2_T_x_, (**b**,**c**) Morphology of PAN/Ti_3_C_2_T_x_ NC, (**d**) Structure of PAN/Ti_3_C_2_T_x_ NC through XRD Analysis, (**e**,**f**) XPS of PANI and PAN/Ti_3_C_2_T_x_ NC for chemical element analysis. Reprinted with permission from ref. [52] Copyright 2020 Elsevier.

**Figure 9 nanomaterials-11-02496-f009:**
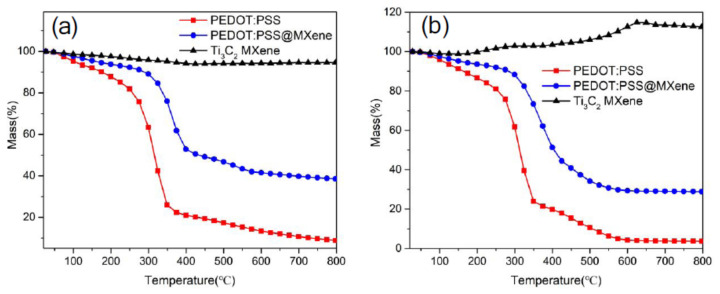
Comparative TGA analysis of MXP-NC with its precursors in (**a**) argon atmosphere, and (**b**) ambient Atmosphere. Reprinted with permission from ref. [65]. Copyright 2020 American Chemical Society.

**Figure 10 nanomaterials-11-02496-f010:**
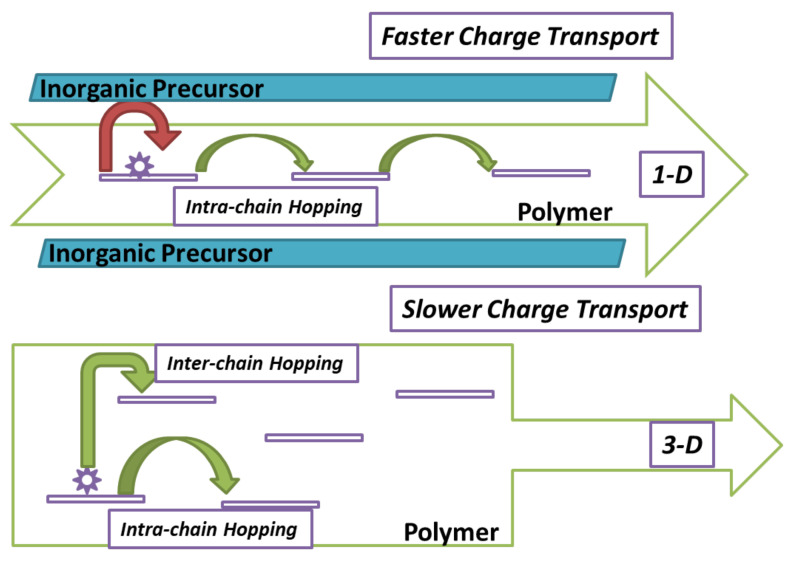
Hoping of charge carrier along 1D in conducting polymers due to addition of inorganic nanomaterials.

**Figure 11 nanomaterials-11-02496-f011:**
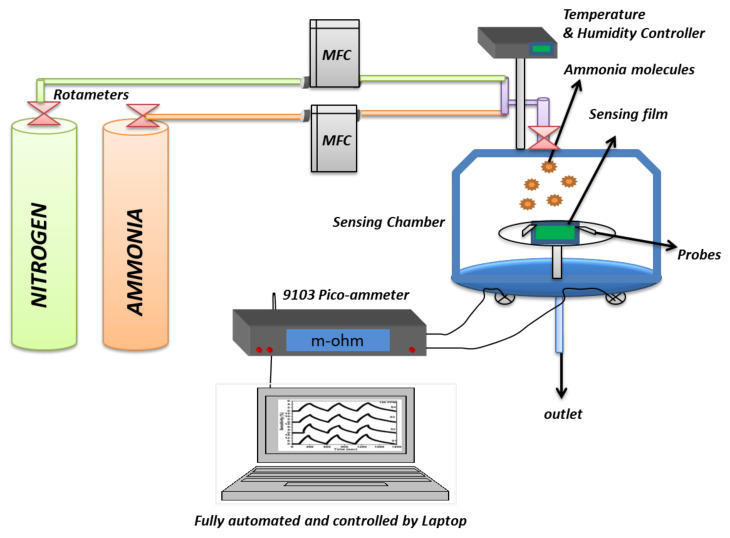
Schematic of ammonia sensing set-up. Reprinted with permission from ref. [69]. Copyright 2021 Wiley.

**Figure 12 nanomaterials-11-02496-f012:**
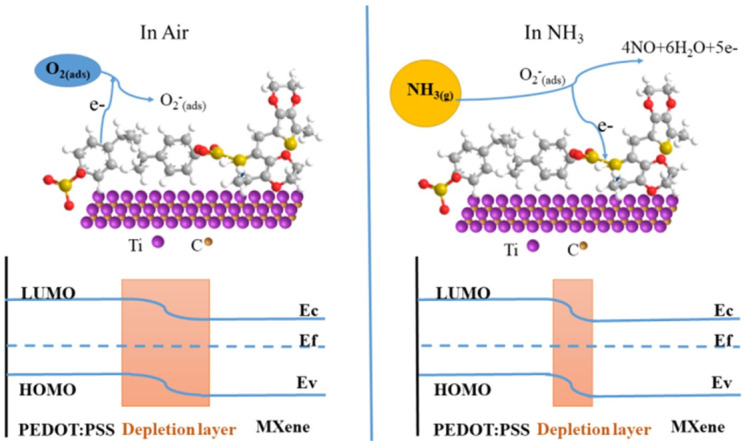
Chemisorption of ammonia in Ti_3_C_2_T_x_/PEDOT: PSS sensors: Reprinted with permission ref. [65]. Copyright 2020 American Chemical Society.

**Figure 13 nanomaterials-11-02496-f013:**
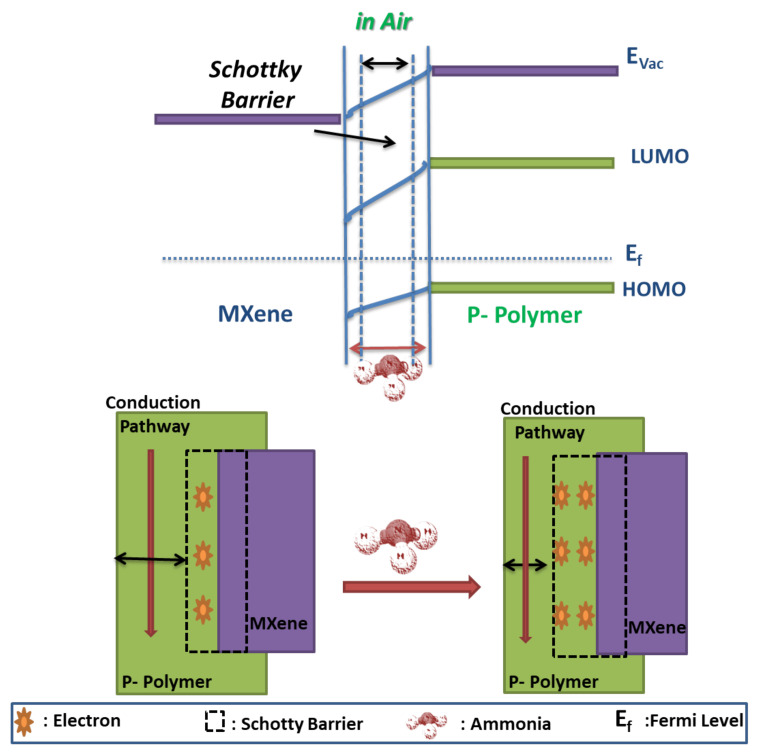
Ammonia-sensing mechanism in p–p-type MXP-NCs due to formation of Schottky barrier.

**Figure 14 nanomaterials-11-02496-f014:**
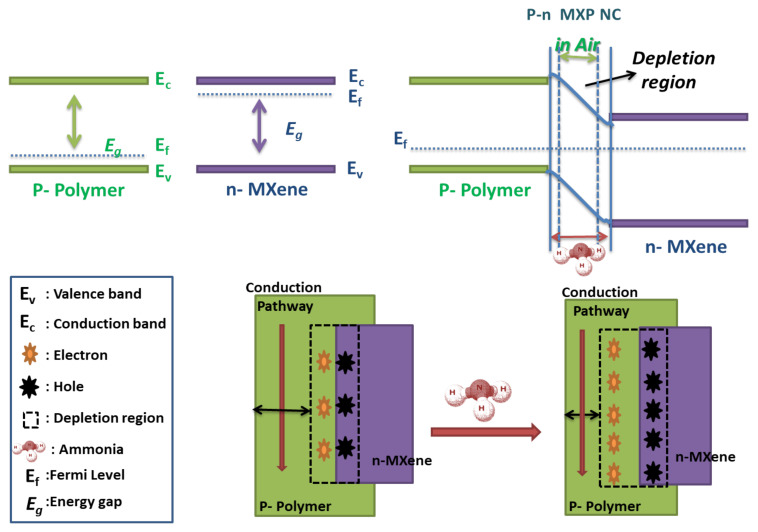
Ammonia-sensing mechanism in p–n-type MXP-NCs due to formation of p–n junctions.

**Figure 15 nanomaterials-11-02496-f015:**
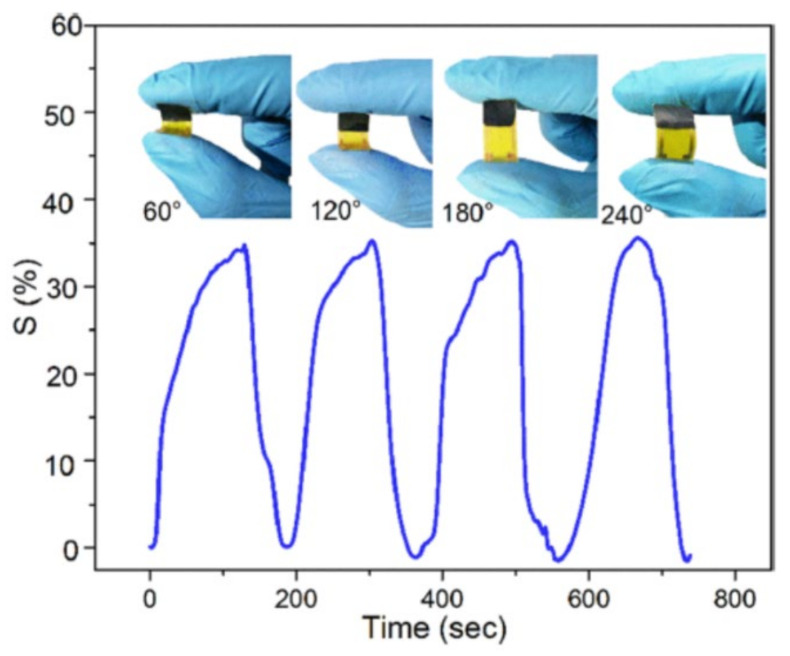
Flexibility of PEDOT: PSS/Ti_3_C_2_T_x_ NC ammonia sensor under different bending angles. Reprinted with permission from ref. [65]. Copyright 2020 American Chemical Society.

**Figure 16 nanomaterials-11-02496-f016:**
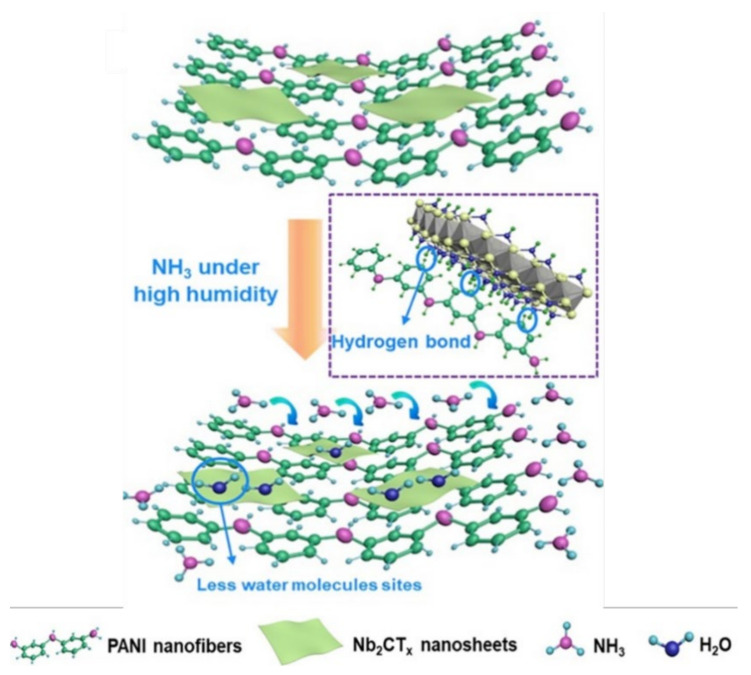
Ammonia-sensing in high humidity conditions. Reprinted with permission from ref. [56] Copyright 2021 Elsevier.

**Figure 17 nanomaterials-11-02496-f017:**
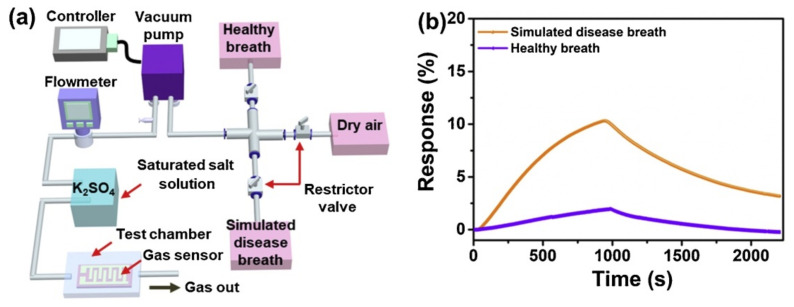
(**a**) Schematic of human breath analysis simulator, (**b**) monitoring of ammonia in exhaled human breath using MXP-NCs. Reprinted with permission from ref. [57]. Copyright 2021 Elsevi.er.

**Figure 18 nanomaterials-11-02496-f018:**
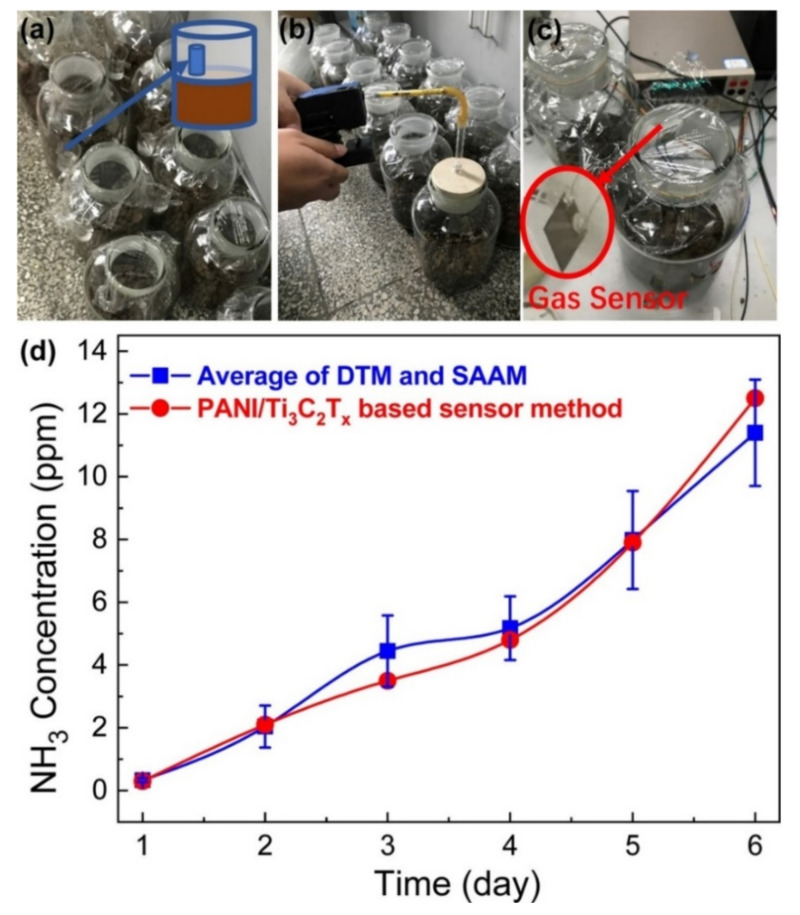
(**a-c**) Set-up to monitoring the volatilization of ammonia using MXP-NCs and (**d**) comparison of MXP-NCs performance in detecting volatilizing ammonia with results from DTM and SAAM techniques. Reprinted with permission from ref. [52]. Copyright 2020 Elsevier.

**Figure 19 nanomaterials-11-02496-f019:**
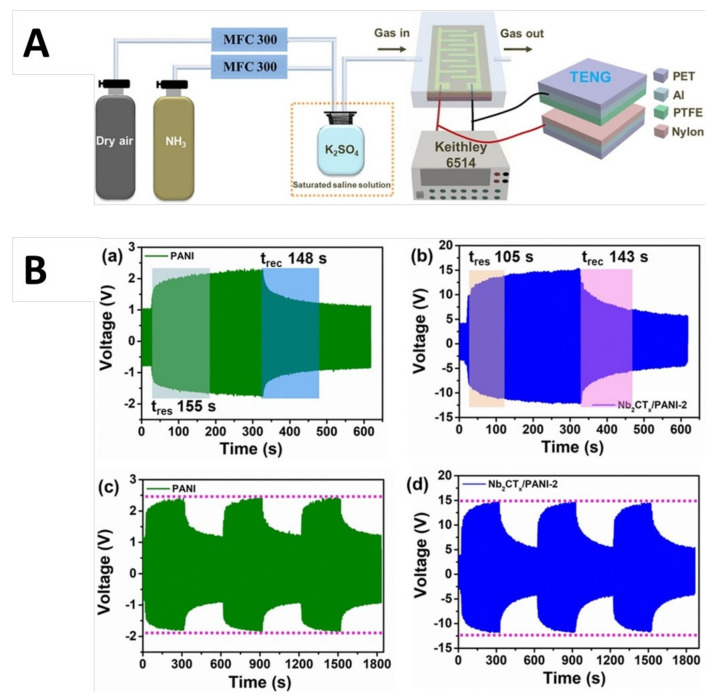
(**A**) Schematic of TENG coupled Nb_2_CT_x_/PAN ammonia sensor and (**B**) External load voltage as sensing parameter for (a,c) Polyaniline sensor, and (b,d) Nb_2_CT_x_/PAN sensor. Reprinted with permission from ref. [56]. Copyright 2021 Elsevier.

**Figure 20 nanomaterials-11-02496-f020:**
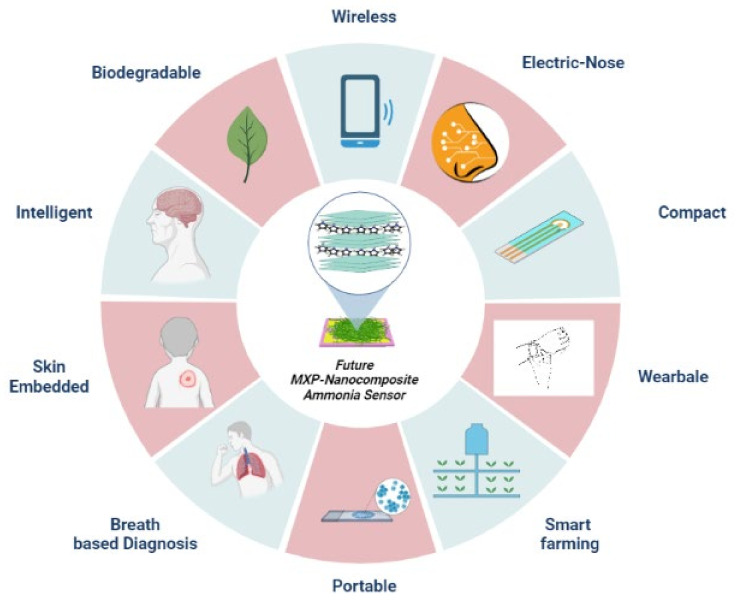
Technological and sustainable prospects of MXP-based ammonia sensor.

**Table 1 nanomaterials-11-02496-t001:** Various types of NH_3_ sensors with respective advantages and disadvantages.

Type of Sensors	Advantages	Disadvantages	Applications
Electrochemical Sensors [10,11,17]	Low detection limit	Short lives	Typical laboratory analysis
	Insensitive to environment change	Malfunction of Electrodes	
	High and accurate sensitivity	Requirement of special design for selective detection	Workplaces such as chemical industries
Spectroscopic Sensors [10,11,17]	Long lifetime	Difficulty in miniaturization to be installed at every emission site	Remote air quality monitoring
		Expensive	
	Insensitive to environment change	Complicated design	Gas leak detection systems with accuracy and safety
	High sensitivity, selectivity, and stability	Time consuming	
Calorimetric Sensors [10,11,17]	Cost effective	Selectivity	Most combustible gases under industrial environment
	Adequate sensitivity for industrial detection	Low versatility	
	Stable at ambient temperature	Risk of catalyst poisoning and explosion	Petrochemical plants
Mass sensitive sensors [10,11,17]	Long lifetime	Low sensitivity	Components of Wireless Sensor Networks.
		Sensitive to environmental change	
	Avoiding secondary pollution	Selectivity	
		Sophisticated equipment	
FET and Diode based Sensors [10,11,17]	Adequate sensitivity	Fabrication complexity	Industrial applications and civil use
	Cost effective	Miniaturization	
	Detailed Analysis	Characterization	
Chemiresistors [10,11,17,18]	Cost effective	Sensitiveness to surrounding environment	Humidity monitoring
	Long lifetime		
	Eco-friendly		Industrial applications
	Energy efficient		

**Table 2 nanomaterials-11-02496-t002:** Comparison of Sensitivity of MXP-NCs with parent constituents and other materials toward NH_3_.

Sensing Material	Sensing Parameter	Sensitivity @ Lowest Detection Range	Temperature of Detection	Reference
Ti_3_C_2_T_x_/PEDOT:PSS	Resistance	4.94% @ 10 ppm	27 °C	[55]
Ti_3_C_2_T_x_/PAN	Resistance	0.05% @ 25 ppb	RT	[39]
Nb_2_CT_x_/PAN-TENG	Voltage	9.33% @ 1 ppm	RT	[45]
Nb_2_CT_x_/PAN	Resistance	1.19% @ 20 ppb	25 °C	[46]
Ti_3_C_2_T_x_/CPAM	Current	1.5% @ 50 ppm	RT	[51]
Nb_2_CT_x_	Voltage	8.15% @ 100 ppm	25 °C	[45]
PAN	Resistance	11% @ 1 ppm	RT	[77]
V_2_CT_x_	Resistance	1.7% @100 ppm	RT	[78]
PEDOT:PSS: Graphene	Resistance	0.9% @ 5 ppm	RT	[79]
PAP	Resistance	9% @ 10 ppm	25 °C	[2]
Alkalise-Ti_3_C_2_T_x_	Resistance	11% @ 10 ppm	25 °C	[80]
PAN-Ag	Resistance	47.1% @ 1 ppm	27 °C	[58]
PAN-GO	Resistance	21.8% @ 0.5 ppm	20 °C	[81]
PAN-MWCNT	Voltage	10% @ 0.01 ppm	RT	[82]

**Table 3 nanomaterials-11-02496-t003:** Comparison of sensing characteristics of MXP-NCs and other NH_3_ sensors.

MXP-NC	Response Time	Recovery Time	Linear Regression Value	Repeatability	Reference
PEDOT:PSS/Ti_3_C_2_T_x_	116 s for 100 ppm	40 s for 100 ppm	0.957 for 10–100 ppm range	3 cycles	[55]
			0.983 for 100–1000 ppm range		
PAN/Ti_3_C_2_T_x_	~600 s for 25 ppb	~1400 s for 25 ppb	0.997 for 2–10 ppm	4 cycles	[39]
CPAM/Ti_3_C_2_T_x_	12.7 s for 150 ppm	14.6 s for 150 ppm	0–2000 ppm	10 cycles	[51]
Nb_2_CT_x_/PAN-TENG	105 s for 100 ppm	143 s for 100 ppm	0.9655 for 1–100 ppm	3 cycles	[45]
Nb_2_CT_x_/PAN	218 s for 10 ppm	300 for 10 ppm	0.9951 for 0.1–10 ppm	3 cycles	[46]
V_2_CT_x_	105 s for 100 ppm	120 s for 100 ppm	Not mentioned	3 cycles	[78]
PAN/MWCNT-TENG	120 s for 100 ppm	127 s for 100 ppm	0.9928 for 20–100 ppm	3 cycles	[82]
Alkaised-Ti_3_C_2_T_x_	1 s for 100 ppm	201 s for 100 ppm	Not mentioned	5 Cycles	[80]
PAN-Ag	271 s for 5 ppm	575 s for 5 ppm	1–100 ppm	4 cycles	[58]

**Table 4 nanomaterials-11-02496-t004:** Summary of selectivity demonstrated by MXP-NCs-based NH_3_ sensors.

Analytes	PEDOT:PSS/Ti_3_C_2_T_x_ [55]	PAN/Ti_3_C_2_T_x_ [39]	CPAM/Ti_3_C_2_T_x_ [51]	Nb_2_CT_x_/PAN-TENG [45]	Nb_2_CT_x_/PAN [46]
Ammonia	36.6% @100 ppm	~1.7% @10 ppm	4.7% @200 ppm	~59% @10 ppm	74.46% @10 ppm
Toluene	1.2% @100 ppm	-	-	-	-
Methanol	14% @100 ppm	-	~15% @2000 ppm	-	-
Ethanol	4.6% @100 ppm	-	~10% @2000 ppm	~3% @10 ppm	~2% @10 ppm
Acetone	3.4% @100 ppm	-	~10% @2000 ppm	~2% @10 ppm	~2% @10 ppm
Sulfur dioxide	-	~0.02% @25 ppm	-	~6% @10 ppm	~9% @10 ppm
Hydrogen Sulfide	-	~1% @25 ppm	-	~2% @10 ppm	~1% @10 ppm
Formaldehyde	-	~0.02% @25 ppm	-	~0.5% @10 ppm	~0.5% @10 ppm
Carbon Monoxide	-	~0.05% @25 ppm	-	~1% @10 ppm	Negligible @10 ppm
Methane	-		-	~2.5% @10 ppm	

**Table 5 nanomaterials-11-02496-t005:** Flexibility feature demonstrated by MXP-NCs-based NH_3_ sensors.

MXP-NC	Bending Angle/Number of Folds				
PEDOT: PSS/Ti_3_C_2_T_x_ over PI substrate: [55]					
Bending Angle (in degrees)	0	60	120	180	240
Sensitivity @100 ppm	36.4%	~35–37%			
PAN/Ti_3_C_2_T_x_ over PI substrate: [39]					
Bending Angle (in degrees)	0	20	30	40	
Bending Cycles: (in number)	0	100	300	500	
Sensitivity @10 ppm	~1.6%				
CPAM/Ti_3_C_2_T_x_ over PET substrate: [51]					
Bending Angle (in degrees)	0	40	60	80	100
Bending Cycles: (in number)	0	3			
Sensitivity @2000 ppm	~45%	~43–45%			

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
