# Peer review of "Emerging MXene–Polymer Hybrid Nanocomposites for High-Performance Ammonia Sensing and Monitoring"

_nanomaterials, 2021, doi:10.3390/nano11102496_

Round 1
Reviewer 1 Report
Some English Corrections are needed: (Please check carefully the whole document)
1. "the maximum exposure limit of NH3 at 25 ppm is 8h, and 35 ppm is 10 min at the workplace [3]."
should be:
the maximum exposure limit of NH3 at 25 ppm is 8h, and when reaching 35 ppm is 10 min at the workplace [3].
2. odema should be edema
3. I do not consider Figure 1 to be appropriate or to bring something new in a journal with FI>5. I suggest cutting it.
4. Table 1 is too generic. It lacks from the point of view of details and of references. Low detection limit (? values-range of values). Please, find some criteria and provide sufficient details and the references are requested.
5. Reformulate: "A promising, demanding, and demanding field"
6. MXenes are not appropriately presented. The significance of notations are not given. Is T , as expected from functional groups and who are the functional groups? Here the authors are lacking with the demanded explanations to be able to discuss further what materials modifications are producing the developments. Any aspect regarding the structure must be discuss and correctly identified to be completely understandable.
7. I suggest here a Table; type of Mxenes; possible functionality and hybrid development; main characteristics; performance regarding sensing;....references
8. Figure 5 is again without major contribution. A phrase can explain all these.
9. Is Figure 10 correct?
10. Figure 11 is not a must.
11. Mechanism should be revised and classification has to be accurate.
12. A review does not mean to publish previously published figures/images, but to develop a new vision based on what is already published and to present this new approach in a newly created figure. So, Figure 18 is again from my point of view without any importance.
13. Figure 20 is a mix of repeated different aspects. My suggestion is to remove it.
14. References need uniformity.
Author Response
Response to the Reviewers Comments
Manuscript ID: nanomaterials-1347922
Type of manuscript: Review
Title: Emerging MXene-Polymer hybrid nanocomposites for high-performance
ammonia sensing and monitoring
Authors: Vishal Chaudhary, Akash Gautam, Yogendra K. Mishra, Ajeet Kaushik *
#Reviewer 1:
Comment: Some English Corrections are needed: (Please check carefully the whole document)
Response: A careful proof-reading has been performed carefully.
Comment 1. "the maximum exposure limit of NH3 at 25 ppm is 8h, and 35 ppm is 10 min at the workplace [3]."
should be:
the maximum exposure limit of NH3 at 25 ppm is 8h, and when reaching 35 ppm is 10 min at the workplace [3].
Response: Suggestion have been incorporated at the required place in manuscript.
Comment 2. odema should be edema.
Response: Correction have been incorporated at appropriate place in manuscript.
Comment 3. I do not consider Figure 1 to be appropriate or to bring something new in a journal with FI>5. I suggest cutting it.
Response: It represents an overview of various strategies, materials and application related ammonia sensing, basically indicating need for and importance of ammonia sensing.
Comment 4. Table 1 is too generic. It lacks from the point of view of details and of references. Low detection limit (? values-range of values). Please, find some criteria and provide sufficient details and the references are requested.
Response: These are very common features of all sensing strategies mentioned in almost every review related to it. The specificity such as ranges and values for strategies depends upon various materials and design. Thus, to specify such things is not possible. However, appropriate references have been added to table.
Comment 5. Reformulate: "A promising, demanding, and demanding field"
Response: Statement has been modified at the relevant place in the manuscript.
Comment 6. MXenes are not appropriately presented. The significance of notations are not given. Is T , as expected from functional groups and who are the functional groups? Here the authors are lacking with the demanded explanations to be able to discuss further what materials modifications are producing the developments. Any aspect regarding the structure must be discuss and correctly identified to be completely understandable.
Response: It has already been included in manuscript:
Various surface functionalities (including hydroxyl, oxygen, fluorine, and chlorine) are represented by 'T' in the formula, resulting from different fabrication approaches [23].
Comment 7. I suggest here a Table; type of Mxenes; possible functionality and hybrid development; main characteristics; performance regarding sensing;....references
Response: It has already been included in various sections related to sensing parameters.
Comment 8. Figure 5 is again without major contribution. A phrase can explain all these.
Answer: It shows two major techniques physical and chemical process for synthesis of MXenes.
Comment 9. Is Figure 10 correct?
Response: Yes, a similar observation has already been reported by one of the authors in reference [79].
Comment 10. Figure 11 is not a must.
Response: It is showing the sensing strategy to detect ammonia.
Comment 11. Mechanism should be revised and classification has to be accurate.
Response: Mechanism has been classified and revised as per suggestion at required place in manuscript.
Comment 12. A review does not mean to publish previously published figures/images, but to develop a new vision based on what is already published and to present this new approach in a newly created figure. So, Figure 18 is again from my point of view without any importance.
Response: Figure 18 illustrates a new application of MXP ammonia sensors in monitoring agricultural ammonia. It is essential from reader’s point of view for better understanding of this different published application.
Comment 13. Figure 20 is a mix of repeated different aspects. My suggestion is to remove it.
Response: It represent future aspects of ammonia sensors based on MXPs.
Comment 14. References need uniformity.
Response: References has been revised as per suggestion in uniform format.

Reviewer 2 Report
This review is well written. In my opinion, it can be published after minor revision.
It is better for the authors to show a sketch of M3X2Tx structure on page 7.
- As for Introduction section, the author lists some 2D materials and polymer materials, and points out that the advantages of MXene-polymer composites are not strong enough. It should be pointed out whether other 2D materials (graphene, black phosphorus, molybdenum disulfide, etc.) -polymer composites are applied in NH3 sensing, and compared with MXene-polymer composites, so as to highlight the advantages of this composite material.
- The full name should be supplied before the abbreviations such as MXP NCs, PVA and PSS.
- In this paper, MXene polymer composites are reviewed, and the authors only supply some Ti3C2Tx-based polymer composite examples, which is only one kind of MXene. The author should add some other kind of MXene-based polymer composite for NH3 detecting and compare their sensing performance.
- MXP-NCs as gas-sensing materials are of certain value to improve the performance of gas sensors towards NH3. However, how to solve the bottlenecks such as the humidity affect the sensing-performance of MXP-NCs sensor? Please give the detailed explanation and evidences in the review.
Author Response
Comment 1 This review is well written. In my opinion, it can be published after minor revision.
It is better for the authors to show a sketch of M3X2Tx structure on page 7.
As for Introduction section, the author lists some 2D materials and polymer materials, and points out that the advantages of MXene-polymer composites are not strong enough. It should be pointed out whether other 2D materials (graphene, black phosphorus, molybdenum disulfide, etc.) -polymer composites are applied in NH3 sensing, and compared with MXene-polymer composites, so as to highlight the advantages of this composite material.
Response: A discussion has been included as per reviewer’s suggestion.
Comment 2: The full name should be supplied before the abbreviations such as MXP NCs, PVA and PSS.
Response: Full form of all the abbreviations stated in the comment has already been mentioned at Page-5 of the manuscript.
Comment 3: In this paper, MXene polymer composites are reviewed, and the authors only supply some Ti3C2Tx-based polymer composite examples, which is only one kind of MXene. The author should add some other kind of MXene-based polymer composite for NH3 detecting and compare their sensing performance.
Response: Other MXene-polymer nanocomposite such as Nb2CTx/PAN, V2CTx based ammonia sensors available in literature, has already been included in the manuscript. (Table- 2-4).
Comment 4: MXP-NCs as gas-sensing materials are of certain value to improve the performance of gas sensors towards NH3. However, how to solve the bottlenecks such as the humidity affect the sensing-performance of the MXP-NCs sensor? Please give the detailed explanation and evidence in the review.
Response: The discussion has already been included in section 8.2.

Round 2
Reviewer 1 Report
The manuscript was significantly improved, but my concerns about new interpretations to literature (especially new drawn figures) to present the interpretations of the Authors of review based on several findings/reports had no success.